# CLIP Behaves like a Bag-of-Words Model Cross-modally but not Uni-modally

**Darina Koishigarina**,* **Arnas Uselis & Seong Joon Oh**
Tübingen AI Center, University of Tübingen

## Abstract

CLIP (Contrastive Language-Image Pretraining) has become a popular choice for various downstream tasks. However, recent studies have questioned its ability to represent compositional concepts effectively. These works suggest that CLIP often acts like a bag-of-words (BoW) model, interpreting images and text as sets of individual concepts without grasping the structural relationships. In particular, CLIP struggles to correctly bind attributes to their corresponding objects when multiple objects are present in an image or text. In this work, we investigate why CLIP exhibits this BoW-like behavior. Our key finding is that CLIP does not lack binding information. Through linear probing, robustness tests with increasing object counts, and conjunctive search experiments, we show that attribute–object bindings are already encoded within CLIP's text and image embeddings. The weakness lies in the cross-modal alignment, which fails to preserve this information. We show it can be accessed cross-modally with a simple linear transformation to text embeddings. This improves CLIP's attribute-object binding performance and confirms that the information was already encoded unimodally. In practice, this means CLIP-based systems can be enhanced with a lightweight linear layer trained on existing embeddings, avoiding costly encoder retraining. The code is available at `https://github.com/kdariina/CLIP-not-BoW-unimodally`.

## 1 Introduction

Vision-language models (VLMs) like Contrastive Language-Image Pretraining (CLIP) (Radford et al., 2021) have achieved widespread adoption due to their shared embedding space for text and image modalities, enabling strong performance on downstream tasks. However, a fundamental limitation has emerged: CLIP often struggles with compositionality (Thrush et al., 2022), specifically the ability to bind attributes to corresponding objects in complex scenes (Tang et al., 2023; Lewis et al., 2024; Yuksekgonul et al., 2023). Compositionality is essential for VLMs, as it allows models to generalize effectively by combining simpler concepts and understanding their relations.

Recent studies (Yuksekgonul et al., 2023) show that CLIP frequently behaves like a bag-of-words (BoW) model, failing to bind attributes to corresponding objects. For instance, given an image of "an orange square and a blue triangle" as in Fig. 1, CLIP often matches the image to a caption "a blue square and an orange triangle". It is often unable to distinguish the structural difference. We refer to this phenomenon as **BoWness**, indicating the model's treatment of each data point as an unordered set of concepts. The BoWness significantly limits CLIP's compositional understanding. Previous research has evaluated this limitation by jointly considering the image and text embeddings. However, there has been little investigation into the source of the inability. In particular, we do not know whether the BoWness arises (1) from a lack of attribute-object binding information in the individual text and image embeddings or (2) from a mere lack of cross-modal alignment.

Distinguishing these two cases is crucial for accurate diagnosis and effective improvement. Current evaluations typically measure binding in the cross-modal space, so poor performance could be misattributed to missing knowledge rather than misalignment. If the limitation lies in the encoders, retraining is required; if it is in cross-modal alignment, a lightweight adjustment may suffice. Clarifying the source can guide the design of downstream VLMs and adapters, ensuring that improvements target the true limitation and that pre-trained models are used effectively.

---

*Corresponding author: `darina51012@gmail.com`

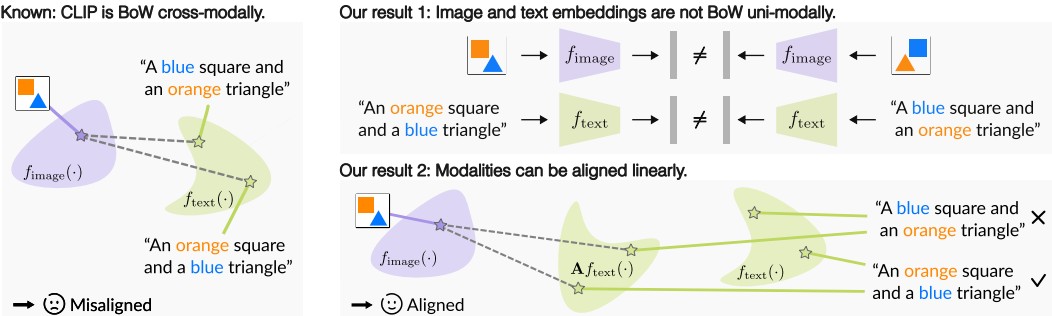

Figure 1: **CLIP is not BoW uni-modally.** (1) It has been reported that CLIP behaves like a BoW model with weak attribute-object binding. (2) We discover that embeddings of individual image and text modalities already contain the attribute-object binding information; this suggests the BoWness stems from the lack of alignment across the modalities. (3) A simple linear transformation of the text modality mitigates the BoWness of CLIP.

In this work, we investigate the cause of CLIP's BoWness by testing whether attribute–object binding is present in individual image and text embeddings. We train linear probes to extract attribute information for specific objects in two-object scenes and show that attributes can be linearly separated in both modalities. This signal remains robust as the number of objects increases, especially for text embeddings. For image embeddings, a conjunctive search experiment confirms that CLIP captures feature bindings, not just a BoW collection of features. Together, these results demonstrate that the embeddings already contain the right attribute-object binding.

Building on this insight, we hypothesize that CLIP's BoW-like behavior stems from superficial cross-modal misalignment. Although binding information is present in each modality, the model is not sufficiently encouraged to use it, leading to mismatched binding signals between text and image embeddings. To validate this, we apply a simple linear transformation $\mathbf{A}$ to one modality. We train $\mathbf{A}$ using negative samples created by permuting attribute-object pairs in text captions from image-caption datasets. Empirically, this approach yields significant gains in cross-modal attribute–object binding on ARO, SugarCrepe, and COCO, indicating that CLIP's encoders were already capable of representing correct bindings. This finding has a practical implication: updating existing CLIP vector databases does not require retraining of the encoders or re-extracting of features.

## 2 RELATED WORK

**Limitations of CLIP's encoders.** Numerous studies have highlighted weaknesses in both CLIP's vision and text encoders. CLIP's visual encoder tends to prioritize high-level understanding, often missing finer details crucial for distinguishing objects (Tong et al., 2024b). Meanwhile, its text encoder struggles with tasks involving negations, spatial and numerical reasoning, and nuanced attribute distinctions (Tong et al., 2024a; Kamath et al., 2023a). These limitations affect performance in downstream applications (Parashar et al., 2024; Tong et al., 2024b;a). Several efforts aim to interpret CLIP's representations to understand these limitations better (Esfandiarpoor et al., 2024; Yun et al., 2023; Bhalla et al., 2024). In contrast, our work specifically focuses on attribute-object binding in scenarios with multiple objects, providing a targeted analysis of its compositional capabilities.

One line of work studies the modality gap, the separation between image and text embeddings. This gap is often viewed as a source of misalignment. However, Schrodi et al. (2024) shows that it arises from the information imbalance between images and text, which limits alignment. Varying the modality gap can affect performance, fairness, and downstream behavior (Schrodi et al., 2024; Liang et al., 2022). We examine whether improving binding changes the modality gap.

**Compositional reasoning and alignment.** Compositional reasoning, critical for understanding complex scenes, has been studied extensively in neural networks (Hupkes et al., 2020; Greff et al., 2020). Prior work on CLIP investigates its ability to handle novel attribute-object combinations (Abbasi et al., 2024; Bao et al., 2023) and attributes its failures to weak compositional reasoning (Lewis et al., 2024; Tang et al., 2023). Some employ controlled setups with two objects and distinct attributes (Lewis et al., 2024; Tang et al., 2023), providing initial insights into the binding problem. Other works specifically study attribute-object binding failures in text-to-image generation models

(Trusca et al., 2024; Zarei et al., 2024). In our work, we ask whether CLIP's embedding space inherently limits binding, and evaluate its capacity to represent binding within and across modalities.

**Benchmarks for compositionality.** A growing number of benchmarks evaluate compositionality in VLMs, often using hard negatives or fine-grained distractors. These include VL-CheckList (Zhao et al., 2022), CREPE (Ma et al., 2023), COLA (Ray et al., 2024), ARO (Yuksekgonul et al., 2023), SugarCrepe (Hsieh et al., 2024), Winoground (Thrush et al., 2022). These benchmarks vary in focus: some test fine-grained distinctions like VisMin (Awal et al., 2024), others use hard positive pairs for reasoning (Kamath et al., 2024), and some target specific challenges like counting in CountBench (Paiss et al., 2023), negation in NegBench (Alhamoud et al., 2025) or spatial reasoning in What'sUp (Kamath et al., 2023b). In contrast, our work specifically focuses on attribute-object binding. Other compositional tasks, such as spatial reasoning or negation, are outside the scope of our study. Synthetic benchmarks like PUG (Bordes et al., 2024) and CLEVR (Johnson et al., 2017) provide controlled environments to test attribute-object binding with targeted scenarios often missing in real-world datasets. Our work extends these efforts by contributing a synthetic, controlled dataset with greater variation, designed to evaluate attribute-object binding.

## 3 THE BINDING PROBLEM

Ideally, VLMs like CLIP need to capture the compositional structure of real-world scenes. A necessary condition for compositional understanding is the ability to accurately bind attributes to corresponding objects in scenes with multiple objects. Prior research suggests that CLIP's attribute-object binding is often arbitrary to the degree that the model can effectively be thought of as a bag-of-words (BoW) extractor that treats objects and attributes in image and text as an unordered collection of concepts, completely ignoring the order and structure therein (Yuksekgonul et al., 2023).

In this work, we define **BoWness** of a vision-language model as the general tendency in models to treat inputs (image or text) as unordered sets of concepts. In contrast, we say that a model has a **binding ability** when it can link attributes correctly to the corresponding objects.

### 3.1 PRELIMINARIES

A CLIP model has two encoders: $f_{\text{image}} : \mathcal{I} \to \mathbb{R}^D$ for images and $f_{\text{text}} : \mathcal{T} \to \mathbb{R}^D$ for texts, where $D$ is the dimensionality of the encoders. For an image $\mathbf{x}^{\text{img}} \in \mathcal{I}$ and a text sequence $\mathbf{x}^{\text{txt}} \in \mathcal{T}$, CLIP embeds both inputs independently into a shared vision-language space. We are interested in the behavior and information content in embeddings $f_{\text{text}}(\mathbf{x}^{\text{txt}})$ and $f_{\text{image}}(\mathbf{x}^{\text{img}})$.

We consider a paired image-text dataset $\mathcal{D} = \{(\mathbf{x}_i^{\text{img}} \mathbf{x}_i^{\text{txt}})\}_{i=1}^N$ of $N$ samples, where each sample consists of a text sequence $\mathbf{x}_i^{\text{txt}} \in \mathcal{T}$ and a corresponding image $\mathbf{x}_i^{\text{img}} \in \mathcal{I}$. Such a dataset typically provides *positive pairs*, where attributes are correctly associated with objects in the text captions, matching the corresponding images. We refer to *negative pairs* as synthetic pairs where the text captions in positive pairs are modified such that the binding is artificially broken through a permutation. For example, given a positive pair with the caption "red cube and blue sphere", we create a negative pair by keeping the image intact and modifying the caption to "blue cube and red sphere".

The creation and usage of negative pairs have been an established strategy in the assessment of CLIP's compositional abilities, as seen in previous benchmarks (Yuksekgonul et al., 2023; Thrush et al., 2022; Hsieh et al., 2024). Some researchers have considered incorporating the negative pairs in training or fine-tuning to equip models with improved compositionality (Yuksekgonul et al., 2023; Patel et al., 2024). Our work, likewise, employs negative pairs for assessing and improving CLIP.

### 3.2 DATASETS

To evaluate CLIP's cross-modal and unimodal binding abilities, we use a mix of real-world and synthetic datasets, as shown below. Further dataset details are in the Appendix A.1.

While real-world benchmarks offer insights into cross-modal alignment, they are too complex for systematically evaluating uni-modal attribute-object binding. Cross-modal binding only requires image-to-text matching, but assessing binding within a single modality requires a controlled set of

Table 1: **Datasets used in our evaluation.** Real-world benchmarks are used for cross-modal binding, while synthetic datasets provide controlled settings for uni-modal binding.

| Dataset | Type | Description / Purpose |
|---|---|---|
| ARO [(Yuksekgonul et al., 2023)] | Real | Tests compositionality with relationships, attributes, and order. |
| SugarCrepe [(Hsieh et al., 2024)] | Real | Comparison against fluent and sensical hard negatives. |
| COCO [(Lin et al., 2014)] | Real | Large dataset for recognition, segmentation, and captioning. |
| CC3M [(Sharma et al., 2018)] | Real | Three million web image-caption pairs for image captioning. |
| CLEVR [(Johnson et al., 2017)] | Synthetic | Simple shapes and colors; controlled environment for binding. |
| PUG:SPAR [(Bordes et al., 2024)] | Synthetic | Animals with controlled attributes; limited by positional bias. |
| PUG:SPARE (ours) | Synthetic | Extension of PUG:SPAR with positional bias removed. |

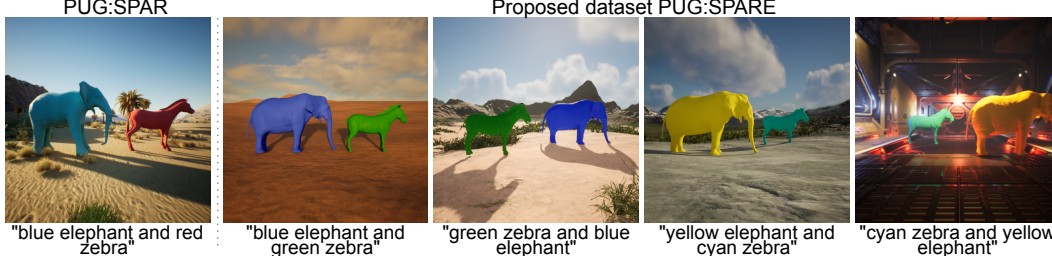

PUG:SPAR — Proposed dataset PUG:SPARE

"blue elephant and red zebra"    "blue elephant and green zebra"    "green zebra and blue elephant"    "yellow elephant and cyan zebra"    "cyan zebra and yellow elephant"

Figure 2: **Examples from PUG:SPAR and PUG:SPARE.** In PUG:SPAR, attributes correlate with object positions: objects on the left are linked to "blue" or "grass" and objects on the right are "red" or "stone". Our dataset PUG:SPARE de-correlates the potential shortcut.

attributes and objects. Real-world datasets lack this control, so we use synthetic datasets with an exact set of objects and attributes, allowing targeted assessments of compositional behavior.

Note that in PUG:SPAR, attributes are tied to positions, which can potentially create shortcuts for models. PUG:SPARE removes this bias by randomizing attributes across positions (see Fig. 2).

We use ARO, SugarCrepe, COCO, and CC3M in §5, and CLEVR, PUG:SPAR, PUG:SPARE in §3.3, 4, 5.

### 3.3 CLIP IS A BAG-OF-WORDS CROSS-MODALLY

In this section, we explain how previous approaches demonstrated the bag-of-words nature of CLIP. We reproduce prior results and confirm that CLIP is BoW cross-modally.

**Previous approach demonstrating BoWness.** A common approach to demonstrate CLIP's BoW-ness behavior is by comparing the embeddings of an image with permutations of its caption (Yuksekgonul et al., 2023). For example, given an image of an orange square and a blue triangle, possible captions could be "an orange square and a blue triangle" and "an orange triangle and a blue square". The task is to identify the correct caption from these options, where one has correct color-object associations and the other has swapped associations. CLIP's prediction is based on which caption embedding has a higher cosine similarity with the image embedding.

Ideally, CLIP should exhibit **cross-modal binding**, or an accurate association of attribute-object pairs across modalities. However, CLIP has been reported to be **cross-modally BoW**, treating the concepts in inputs as an unordered collection. These opposing behaviors are reflected in how cosine similarities rank input pairs, as shown below.

**Cross-modally BoW:**

$$f_{\text{image}}(\blacksquare\blacktriangle)^{\top} f_{\text{text}}(\text{"orange square and blue triangle"}) \approx f_{\text{image}}(\blacksquare\blacktriangle)^{\top} f_{\text{text}}(\text{"blue square and orange triangle"})$$

**Cross-modal binding:**

$$f_{\text{image}}(\blacksquare\blacktriangle)^{\top} f_{\text{text}}(\text{"orange square and blue triangle"}) > f_{\text{image}}(\blacksquare\blacktriangle)^{\top} f_{\text{text}}(\text{"blue square and orange triangle"})$$

**Replicating BoWness results.** Using this approach on the datasets discussed previously, *our results confirm prior findings*. Specifically, when choosing between the two options (correct and permuted), we observe *0.56 accuracy on CLEVR, 0.51 on PUG:SPAR, and 0.50 on PUG:SPARE*, indicating that CLIP's performance is virtually at the level of random guessing. These results strongly suggest that CLIP cannot distinguish between correct and permuted attribute-object bindings. CLIP is indeed a bag-of-words model.

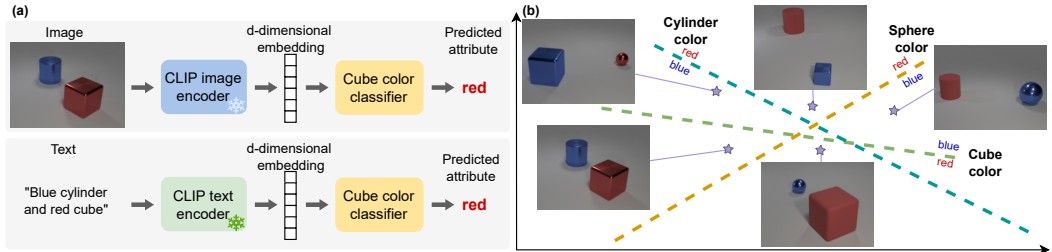

Figure 3: **Uni-modal attribute-object binding.** (a) we train a linear probe per object to distinguish its color within image and text modality separately. (b) linear probes establish decision boundaries in CLIP's representation space that differentiate between various attribute-object associations.

## 4 CLIP BINDS CONCEPTS UNIMODALLY

Prior works evaluating CLIP's BoW tendencies have relied on assessments that combine both text and image modalities. This approach has a key limitation: it does not separate the encoding of attribute-object binding within each modality from the cross-modal matching step. As a result, it remains unclear whether CLIP's BoW behavior stems from limitations in the embeddings themselves or from issues in cross-modal alignment. To address this, we examine the **uni-modal binding**, referring to CLIP's ability to encode attribute-object relationships independently within each modality. By evaluating text and image embeddings separately, we aim to clarify whether each modality alone captures sufficient binding information, shedding light on the roots of CLIP's BoW behavior.

### 4.1 LINEAR PROBING FOR UNI-MODAL BINDING

To evaluate if CLIP encodes binding information within each modality, we use linear probing (Alain, 2016). By training linear classifiers for the information we care about on top of frozen representations, we assess the existence of information we seek in the representation. In this case, we seek the attribute-object binding information in each modality; we train linear classifiers separating attributes for each object present in the image or text inputs. We focus on scenes with two objects, the simplest non-trivial case, where at least two binding configurations are possible.

Given a dataset $\mathcal{D} = \{(\mathbf{x}_i^{\text{img}}, \mathbf{x}_i^{\text{txt}})\}_{i=1}^N$, we define $\mathcal{O}$ as the set of all objects and $\mathcal{A}$ as the set of all attributes in the dataset. For each object $o \in \mathcal{O}$, we train two separate classifiers, for images and for text. These classifiers are trained with embeddings extracted from the respective CLIP encoders, which remain frozen throughout the training. We denote object-specific probes as $\texttt{image-probe}_o$ and $\texttt{text-probe}_o$, each predicting the attribute $a \in \mathcal{A}$ for the object $o$:

$$\texttt{image-probe}_o : f_{\text{image}}(\mathbf{x}_i^{\text{img}}) \mapsto a, \quad \texttt{text-probe}_o \ : f_{\text{text}}(\mathbf{x}_i^{\text{txt}}) \mapsto a. \tag{1}$$

For example, as illustrated in Fig. 3(a), given an image or text describing "blue cylinder and red cube" we extract corresponding embeddings using the CLIP encoders. We then train a linear classifier to recognize the attribute (e.g. $a =$"red") of a specific object (e.g. $o =$"cube"). This process allows us to isolate and probe attribute recognition for each object individually within each modality. This differs from cross-modal retrieval in benchmarks such as ARO and SugarCrepe, where one modality is queried against the other (image-to-text or text-to-image). In such cases, it is difficult to disentangle whether errors arise from the embeddings themselves or from their alignment.

We use synthetic datasets here because they provide the necessary coverage of objects and attributes to train and test probes systematically. In CLEVR and PUG:SPARE, the linear probes classify among 8 possible attribute classes. In PUG:SPAR, they classify among 4 classes. The datasets are split into train, validation, and test sets such that each attribute-object binding pair is unique to a single split. This prevents reliance on memorized bindings and ensures evaluation on unseen compositions. To quantify binding information in each modality, we report the test accuracy of linear probes, averaged across all objects. This corresponds to how linearly separable the attributes are in CLIP's representations, as illustrated in Fig. 3(b).

To contextualize the information content in pre-trained CLIP representations, we provide accuracies of random baseline $(1/|\mathcal{A}|)$ and CLIP encoders fine-tuned for the attribute prediction task. They

provide the reference points for no binding information and maximal binding information in the encoders, respectively. See Appendix A.3 for further details on linear probing.

**Results.** Table 2 presents the classification accuracies for the linear probes on text and image embeddings for CLEVR, PUG:SPAR, and PUG:SPARE. The probes trained on pre-trained CLIP embeddings achieve accuracies far beyond the random baseline for both image (e.g. 0.96 on image test set compared to 0.12 for CLEVR) and text (e.g. 1.00 on text vs to 0.12 for random) modalities. The accuracies are close to the maximal information bound given by the fine-tuned CLIP.

**Sanity check: BoW models lack discriminative signal.** We confirm that a BoW representation lacks the structure necessary for binding. We train randomly initialized CLIP encoders under a BoW constraint, where they recognize all attributes but do not associate them with objects. Both encoders are trained to predict the presence of each attribute using soft label cross-entropy loss.

Table 2: **CLIP is not BoW uni-modally.** Linear probe accuracies on CLIP's image and text embeddings show that binding information is preserved within each modality. Random and fine-tuned encoders provide minimal and maximal reference points, respectively.

| Dataset | Encoder | Image Train | Image Test | Text Train | Text Test |
|---|---|---|---|---|---|
| CLEVR | Random | 0.12 | 0.12 | 0.12 | 0.12 |
| | CLIP | 1.00 | 0.96 | 1.00 | 1.00 |
| | CLIP (ft) | 1.00 | 0.99 | 1.00 | 1.00 |
| PUG:SPAR | Random | 0.25 | 0.25 | 0.25 | 0.25 |
| | CLIP | 1.00 | 0.99 | 1.00 | 0.99 |
| | CLIP (ft) | 1.00 | 0.98 | 1.00 | 1.00 |
| PUG:SPARE | Random | 0.12 | 0.12 | 0.12 | 0.12 |
| | CLIP | 0.99 | 0.95 | 1.00 | 1.00 |
| | CLIP (ft) | 1.00 | 1.00 | 1.00 | 1.00 |

Linear probing on the resulting embeddings yields significantly worse accuracy compared to the original CLIP (0.66 for BoW image embeddings, 0.85 for BoW text embeddings vs. 0.96 for original CLIP embeddings). This confirms that BoW models lack the structure needed for binding. High-dimensional embeddings alone do not guarantee that binding can be recovered with a linear probe.

> **Takeaway §4.1:** CLIP embeddings encode attribute-object binding in a linearly separable way. This separability does not hold cross-modally, suggesting the problem lies in alignment.

## 4.2 UNI-MODAL BINDING WITH MORE OBJECTS

To evaluate the robustness of CLIP's uni-modal attribute-object binding, we extend the experiments in §4.1 by varying the number of objects per scene. We increase the object count within the CLEVR dataset and measure the accuracy of linear classifiers trained to identify object-specific attributes in the representations.

Fig. 4 shows the attribute probing accuracy as a function of object count for text and image modalities. The text modality maintains high accuracy, consistently above 0.8 across all counts, indicating stable uni-modal binding. Image embeddings decline gradually, from about 0.9 with two objects to around 0.6 with many objects. Still, performance stays well above chance, showing that binding signals persist even in cluttered scenes. The discrepancy in performance for the two modalities may stem from how they represent objects. In text, objects are expressed through separate tokens, which keep them distinct. In images, pixels from different objects overlap, so the encoder must disentangle them, making binding harder as scenes grow crowded.

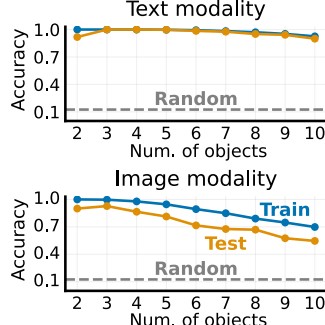

Figure 4: **Image and text embeddings encode multiple objects.** Average linear probing accuracy on CLEVR as the number of objects increases.

> **Takeaway §4.2:** CLIP's text embeddings preserve binding even with many objects. Image embeddings, while challenged by complexity, still retain binding signals above chance.

## 4.3 CONJUNCTIVE SEARCH

In this section, we show that CLIP's visual embeddings encode binding using a visual search experiment. This provides further evidence that CLIP is not a BoW model within the image modality.

We adapt the conjunctive search experiment from Campbell et al. (2024), originally designed to study binding in multimodal language models. Unlike their cross-modal evaluation, we focus on

CLIP's visual encoder to examine its uni-modal binding. Notably, their experiments on LLaVA-1.5 (Liu et al., 2023b), which uses a pre-trained CLIP ViT-L/14 as its image encoder, showed poor performance. Again, this raises the question of whether the limitation stems from the visual embeddings or the interaction between the two modalities.

**Method.** The conjunctive search task requires identifying a target object among distractors, where the target shares one feature (e.g., color or shape) with each distractor type. As illustrated in Fig. 5, an image contains many green spheres and red cubes. The task is to determine whether a red sphere is present. The red sphere shares color with the red cubes and shape with the green spheres. It has no distinguishing features from other objects except for the unique binding of the features. Half of the images include the red sphere (incongruent case), while the other half do not (congruent case).

To test CLIP, we trained binary linear classifiers on its visual embeddings to predict whether an image contained an incongruent object. Performance is evaluated on a test set as the number of distractor objects increases. We also conducted a zero-shot classification using captions and included a randomly initialized CLIP model for comparison. More details are in Appendix A.4.

**Results.** The results (Fig. 5, right) show that pre-trained CLIP embeddings enable accurate binary classification of incongruent objects, regardless of cluttered scenes (accuracy exceeds 0.80 for up to 35 objects in both settings). In contrast, zero-shot classification and the randomly initialized model stay at the random baseline level, showing no meaningful signal for binding.

These results suggest that, while CLIP fails to align captions to images in zero-shot settings, its visual embeddings are not purely BoW. If they were, the embeddings would reflect frequent concepts ("green", "sphere", "red", and "cube") without significant change when a red sphere is added. However, the linear clas-

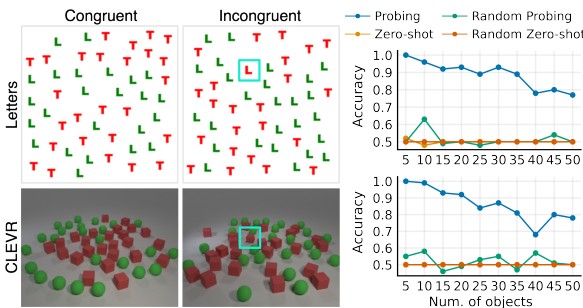

Figure 5: **CLIP embeddings identify objects with shared features but unique bindings.** *Left*: Examples from the letters and CLEVR settings (Campbell et al., 2024). Incongruent examples contain an object sharing one feature (color or shape) with others but with different binding. Congruent examples lack this extra object. *Right*: Linear and zero-shot classification accuracies on congruent and incongruent cases.

sifier's success in distinguishing incongruent cases indicates that CLIP's embeddings encode a nuanced binding between attribute "red" and object "sphere", enabling the recognition.

> **Takeaway §4.3:** CLIP's visual embeddings support conjunctive search, allowing detection of the object defined only by a unique attribute-object binding, even in cluttered scenes.

We conclude that CLIP is already aware of attribute-object bindings in individual modalities. We hypothesize that CLIP's BoWness stems from the poor cross-modal alignment that takes place after the encoding. We verify the hypothesis in the next section.

## 5 IMPROVING CROSS-MODAL BINDING

We observed that CLIP is not a bag-of-words (BoW) model uni-modally. We thus narrow down the root cause of the previously observed cross-modal BoWness to a poor cross-modal alignment in the representation space. In this section, we verify this by proposing a simple alignment strategy that recovers the attribute-object binding information across the modalities. Specifically, we apply a linear transformation to one modality's embeddings (e.g., text) to ensure cosine similarities retrieve pairs with correct binding first. We refer to this method as LABCLIP.

### 5.1 METHOD

Our linear probing experiments revealed that both vision and text spaces encode attribute-object information in a linearly separable manner. This suggests that each modality organizes information in a way that could simplify alignment. If the internal structures of the two spaces are linearly separable, a linear transformation could map these to a shared alignment, improving cross-modal cor-

respondence. CLIP's contrastive loss encourages global alignment but does not focus on aligning attribute-object binding. The original alignment mechanism does not fully use the binding information, which is linearly accessible within each modality. To address this, we propose learning a linear transformation between the text and image spaces using pre-trained CLIP embeddings.

Linear Attribute Binding CLIP (LABCLIP) trains a linear transformation to better align the text and image embeddings. Instead of the standard image-text matching in CLIP,

$$\langle f_{\text{image}}(\mathbf{x}^{\text{img}}), f_{\text{text}}(\mathbf{x}^{\text{txt}})\rangle, \tag{2}$$

LABCLIP applies a matrix $\mathbf{A} \in \mathbb{R}^{D \times D}$ on the text embeddings before the inner product:

$$\langle f_{\text{image}}(\mathbf{x}^{\text{img}}), \mathbf{A} f_{\text{text}}(\mathbf{x}^{\text{txt}})\rangle. \tag{3}$$

Applying the transformation to text or image embeddings is mathematically equivalent. We apply it to text embeddings for practical reasons. Keeping the visual encoder unchanged preserves its utility in downstream tasks, where the original CLIP image embeddings are often used. Additionally, negative texts are easier to obtain than negative images. Experiments aligning the image space confirm that the results remain consistent (see Table 9 in the Appendix).

**Training**. We train the linear transformation matrix $\mathbf{A}$ contrastively, initialized from the identity matrix, with no further constraints. CLIP's weights remain frozen. We include negative text samples within the batch, similar to (Yuksekgonul et al., 2023). Negative samples are created by permuting the concepts in the original captions: we transform "red cube and blue sphere" into "blue cube and red sphere" without changing the image. Such samples can be generated without extra annotation costs. The captions in COCO and CC3M are not structured in a straightforward "attribute-object" format, making it challenging to isolate and swap attributes and objects directly. Because of this, we use the NegCLIP strategy from (Yuksekgonul et al., 2023), which shuffles nouns and adjectives to create negative samples. In training, including negative text samples results in a $B \times 2B$ batch, where the transformation is used to minimize the similarity with mismatched attribute-object pairs.

See Appendix A.5 for the details of the experimental setup and Appendix A.6 for additional analyses of the effect of LABCLIP on the CLIP's shared embedding space.

> **Takeaway §5.1:** LABCLIP does not update CLIP's parameters. It intends to leverage existing binding signals and aligns them through a linear transformation. This approach is efficient to train and backward compatible with existing CLIP-based vector database systems.

## 5.2 RESULTS

Table 3 provides the cross-modal binding results on CLEVR, PUG:SPAR, and PUG:SPARE. The first column is the accuracy of a model in matching an image to the correct caption or the permuted caption. Recall@1 measures the model's ability to retrieve the correct caption from all possible captions in the dataset. For a reference point, we also provide results with a fine-tuned CLIP model (CLIP-FT).

Without training, the original CLIP results are at the random chance level, as discussed in Section 3.3: CLIP cannot differentiate between correct and permuted captions cross-modally. Fine-tuning with negative samples enables near-perfect scores, providing an upper bound on the possible

Table 3: **LABCLIP recovers cross-modal binding.** We measure the ability to rank attribute–object pairs correctly. "Frozen?" indicates whether CLIP's encoders are updated. "#Params" reports the number of learnable parameters. While baseline CLIP performs near random, LABCLIP shows near upper-bound performance of fine-tuned CLIP.

| Dataset | Model | Frozen? | #Params | Accuracy Train | Accuracy Test | Recall@1 Train | Recall@1 Test |
|---|---|---|---|---|---|---|---|
| CLEVR | Random | – | – | 0.50 | 0.50 | 0.01 | 0.01 |
| | CLIP | – | – | 0.49 | 0.58 | 0.25 | 0.36 |
| | LABCLIP | ✓ | 589.8K | 1.00 | 0.95 | 1.00 | 0.93 |
| | CLIP-FT | ✗ | 428.2M | 1.00 | 1.00 | 0.99 | 0.97 |
| PUG:SPAR | Random | – | – | 0.50 | 0.50 | 0.00 | 0.00 |
| | CLIP | – | – | 0.52 | 0.53 | 0.08 | 0.09 |
| | LABCLIP | ✓ | 589.8K | 1.00 | 0.97 | 0.98 | 0.91 |
| | CLIP-FT | ✗ | 428.2M | 1.00 | 1.00 | 1.00 | 0.99 |
| PUG:SPARE | Random | – | – | 0.50 | 0.50 | 0.00 | 0.00 |
| | CLIP | – | – | 0.50 | 0.50 | 0.06 | 0.06 |
| | LABCLIP | ✓ | 589.8K | 0.98 | 0.94 | 0.95 | 0.90 |
| | CLIP-FT | ✗ | 428.2M | 1.00 | 1.00 | 1.00 | 1.00 |

attribute-object binding. We observe that LABCLIP improves performance compared to CLIP. On CLEVR, for example, it achieves an accuracy of 0.95, significantly greater than CLIP's 0.58.

Note that our CLEVR results differ from the two-object experiment in Lewis et al. (2024). Their evaluation tests both binding and compositional generalization, while we focus only on attribute-object binding. See Appendix A.5 for details on the differences.

Table 4: **LABCLIP enhances compositional reasoning on real-world benchmarks.** We compare the performance to baseline CLIP and fine-tuned CLIP with negative examples (NegCLIP). CLIP and NegCLIP update all parameters of the ViT-B/32 backbone (151M learnable parameters). LABCLIP adds only a lightweight $512 \times 512$ linear layer (262K learnable parameters) on top of the frozen CLIP encoders.

| Model | Dataset | Frozen? | ARO | | | | SugarCrepe | | | COCO |
|---|---|---|---|---|---|---|---|---|---|---|
| | | | VG-A | VG-R | Flickr PRC | COCO PRC | Add | Replace | Swap | Recall@1 |
| CLIP | – | – | 0.63 | 0.63 | 0.60 | 0.48 | 0.73 | 0.80 | 0.62 | 0.30 |
| NegCLIP | COCO | ✗ | 0.71 | 0.81 | 0.91 | 0.86 | 0.87 | 0.85 | 0.75 | 0.41 |
| LABCLIP | COCO | ✓ | 0.69 | 0.82 | 0.84 | 0.81 | 0.81 | 0.82 | 0.74 | 0.41 |
| LABCLIP | CC3M | ✓ | 0.68 | 0.78 | 0.83 | 0.73 | 0.83 | 0.80 | 0.68 | 0.34 |

Table 4 shows the results on the real-world benchmarks, ARO (Yuksekgonul et al., 2023) and SugarCrepe (Hsieh et al., 2024), when trained on COCO (Lin et al., 2014) and CC3M (Sharma et al., 2018). LABCLIP significantly outperforms the standard CLIP, indicating a better understanding of attributes, relations, and word order. LABCLIP trained on CC3M performs slightly below the COCO-trained version on some compositional benchmarks, which is expected given the closer match between COCO and the ARO and SugarCrepe datasets (Oh et al., 2024). Importantly, LABCLIP-CC3M still shows strong gains over CLIP, suggesting that the performance improvements stem from improved object-attribute binding, not from dataset similarity.

We present these results alongside NegCLIP, a fine-tuned CLIP model trained with negative image–text pairs from COCO (Yuksekgonul et al., 2023). NegCLIP achieves strong results, but LABCLIP is not intended to outperform it or other state-of-the-art methods. Instead, LABCLIP's results demonstrate that since CLIP embeddings are not BoW uni-modally and samples with permuted bindings are linearly separable, this structure can be leveraged through a linear transformation.

Despite its simplicity, LABCLIP matches NegCLIP on compositional benchmarks, while offering practical benefits. Rather than retraining vision or text encoders to improve binding and re-extracting all features, LABCLIP only requires training a lightweight linear layer on top of existing text embeddings. This means that it can work directly with existing CLIP vector databases. This offers backward compatibility with deployed CLIP systems, post hoc modularity without altering the pre-training pipeline, and far greater efficiency (training is over $100\times$ faster than NegCLIP on CLEVR).

We also test LABCLIP on a spatial reasoning dataset (Appendix A.5). LABCLIP matches CLIP's spatial performance and improves when trained with spatial data.

**Downstream performance**. We report downstream results in Table 11 in the Appendix. LABCLIP performs slightly worse than CLIP on single-object classification tasks (CIFAR, ImageNet), suggesting a tradeoff between coarse object-level signals and attribute-object binding. One possible explanation is that binding supervision shapes the linear transformation toward a binding-specific structure rather than broad object discrimination.

Our results suggest that better cross-modal binding can be achieved by linearly transforming one of the embedding spaces without requiring extensive computations, complex methodologies, or *any change to CLIP parameters*. This further corroborates our previous findings that all the ingredients and information for attribute-object binding are already present in the pre-trained CLIP models.

> **Takeaway §5.2:** A simple linear transformation recovers available binding information, thereby improving cross-modal binding both on synthetic and real-world benchmarks.

## 5.3 THE EFFECT OF ALIGNMENT ON LINEAR PROBING

Training CLIP enables the image and text encoders to learn attribute-object binding within their own modalities, as shown in our uni-modal binding experiments. However, this binding is not aligned across modalities. According to (Yuksekgonul et al., 2023), a bag-of-words strategy suffices for CLIP's high performance. Moreover, the VLMs tend to be biased towards objects rather than attributes because objects appear more often in captions (Schrodi et al., 2024). We hypothesize that because CLIP's training objective does not depend on attribute-object binding, there is no incentive to align binding signals across modalities. While the encoders learn attribute-object binding, they fail to align these signals effectively.

**Method.** To investigate this misalignment, we compare the coefficients of linear probes for image and text embeddings before and after alignment. We first compute image embeddings $f_{\text{image}}(\mathbf{x}^{\text{img}})$ and text embeddings $f_{\text{text}}(\mathbf{x}^{\text{txt}})$. Then, we obtain aligned text embeddings $\mathbf{A}\,f_{\text{text}}(\mathbf{x}^{\text{txt}})$ by multiplying the alignment matrix with the text embeddings. We normalize all the embeddings and learn object-specific probes as described in §4.1. We define $\mathbf{w}_{\text{img}}, \mathbf{w}_{\text{txt}}, \mathbf{w}_{\text{aligned-txt}} \in \mathbb{R}^{D \times |\mathcal{A}| \times |\mathcal{O}|}$ as concatenated vectors of object-specific linear probe coefficients for image embeddings, original text embeddings, and aligned text embeddings, respectively. We then compute the probe similarities:

$$\texttt{cos-sim}(\mathbf{w}_{\text{img}}, \mathbf{w}_{\text{txt}}) = \frac{\langle \mathbf{w}_{\text{img}}, \mathbf{w}_{\text{txt}} \rangle}{\|\mathbf{w}_{\text{img}}\| \|\mathbf{w}_{\text{txt}}\|}, \quad \texttt{cos-sim}(\mathbf{w}_{\text{img}}, \mathbf{w}_{\text{aligned-txt}}) = \frac{\langle \mathbf{w}_{\text{img}}, \mathbf{w}_{\text{aligned-txt}} \rangle}{\|\mathbf{w}_{\text{img}}\| \|\mathbf{w}_{\text{aligned-txt}}\|}. \quad (4)$$

**Results.** The results are illustrated in Fig. 6. Before alignment, the cosine similarity of probe weights is low (0.20, 0.18, and 0.09 for CLEVR, PUG:SPAR, and PUG:SPARE, respectively), indicating a differing encoding of binding. After aligning text embeddings with image embeddings, the similarities increase significantly to 0.75, 0.78, and 0.65. This increase confirms that the alignment process effectively adjusts the text embeddings to match the attribute-object binding encoded in the image embeddings, whereas CLIP's original alignment mechanism does not naturally support cross-modal binding.

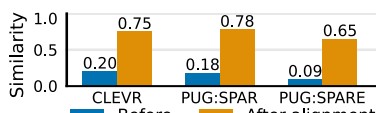

Figure 6: **Alignment enhances similarity between image and text probes.** Cosine similarities of probe coefficients before and after alignment show an increase, confirming alignment of binding across modalities.

> **Takeaway §5.3:** The simple linear transformation on text embeddings aligns the attribute-object binding signals in the image and text modalities.

## 6 CONCLUSION

In this study, we investigated the reasons for CLIP's bag-of-words behavior, focusing on attribute–object binding. We showed that binding information is already present in CLIP's text and image embeddings: it is linearly separable, remains robust as the number of objects increases, and supports conjunctive search. The poor performance, therefore, stems not from missing information but from misalignment. To validate this, we introduced LABCLIP, a simple linear transformation applied to text embeddings. LABCLIP recovers the unimodal binding signals during cross-modal matching, enhancing compositional understanding. It also offers practical advantages: it requires no changes to CLIP encoders, making it efficient to train, modular as a post hoc method, and backward compatible with existing systems. Our work motivates further exploration into the properties of CLIP embeddings uni-modally and into alignment strategies that enhance compositional reasoning.

## 7 ACKNOWLEDGMENTS

We thank the anonymous reviewers for their helpful feedback. This work was supported by the Tübingen AI Center. Arnas Uselis was supported by the International Max Planck Research School for Intelligent Systems (IMPRS-IS).

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

# A  APPENDIX

## A.1  DATASETS

In this section, we provide details for the datasets introduced in Section 3.2. A summary of the key dataset characteristics for the CLEVR, PUG:SPAR, and PUG:SPARE is presented in Table 5.

Table 5: **Specifications for the datasets used to test attribute-object binding in a controlled setting.** For CLEVR, the number of attribute-object combinations only reflects the two-object case. For PUG:SPAR, the numbers represent the filtered dataset used in our experiments.

|                                | CLEVR  | PUG:SPAR       | PUG:SPARE                                    |
|--------------------------------|--------|----------------|----------------------------------------------|
| #images                        | 5000   | 19840          | 88704                                        |
| #attribute-objects combinations| 192    | 1984           | 3696                                         |
| #objects                       | 3      | 32             | 12                                           |
| #attributes                    | 8      | 4              | 8                                            |
| #backgrounds                   | 1      | 10             | 4                                            |
| positions                      | random | {left, right}  | {left, right} × {front, back, equal}         |

### A.1.1  CLEVR

**Generation**.  Following the CLEVR dataset introduced in Johnson et al. (2017), we generate new images using the 3D modeling software Blender Community (2018).  The dataset contains images with $M$ colored objects and corresponding captions.  The set of objects is $\mathcal{O} = \{\text{cube}, \text{sphere}, \text{cylinder}\}$, and the attributes are selected from the set of eight colors:  $\mathcal{A} = \{\text{blue}, \text{red}, \text{purple}, \text{cyan}, \text{gray}, \text{brown}, \text{green}, \text{yellow}\}$.

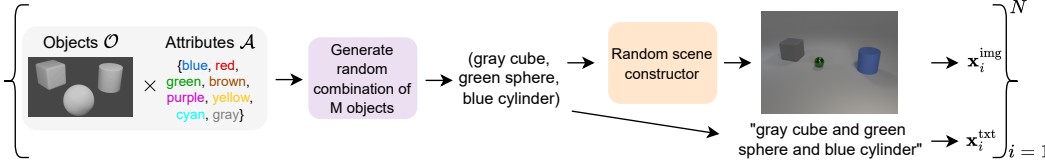

Figure 7: **CLEVR dataset generation process.** From a set of objects and attributes, we randomly generated combinations of $M$ objects per scene. Each combination was rendered with Blender, and a caption was generated by concatenating attributes and objects to match the image.

The data generation process is shown in Fig. 7. We randomly sample objects and attributes to create combinations of $M$ objects of various attributes. For example, an attribute-object combination of 5 objects is 2 blue cubes, 1 green sphere, 1 purple sphere, and 1 blue cylinder. For scenes with two objects, we enforce that the objects are distinct. This results in 192 unique combinations of two objects: 3 choices for the first object, 2 for the second, 8 color options per object (colors can repeat), divided by 2 to ignore object order: $3 \times 2 \times 8 \times 8 \times 0.5 = 192$

Using the original CLEVR image generation pipeline, we construct a scene in Blender based on the sampled combinations. Objects are placed randomly on a neutral background with variations in material and size. The rendered images have dimensions of $320 \times 240$. For captions, we concatenate the attributes and objects in the format: "$a_1\ o_1$ *and* $a_2\ o_2$ *and ... and* $a_M\ o_M$" where $a_j \in \mathcal{A}$ and $o_j \in \mathcal{O}$ for $j \in 1, ..., M$.

We generate $N = 5000$ samples for each $M$-object configuration. For the experiments in Sections 4 and 5, we use $M = 2$. For testing uni-modal binding with an increasing number of objects in Section 4.2, we consider $M$ ranging from 2 to 10. Sample images and captions are shown in Fig. 8.

**Train/test split**.  For each $M$-object setting, we divide the dataset into training, validation, and test sets in a 90/10/10 ratio based on attribute-object combinations. For instance, if there are 192 attribute-object combinations for the two-object setting, 19 combinations are assigned to the validation set, 19 to the test set, and the remaining combinations to the training set. This ensures that the same combination does not appear in both the training and test sets.

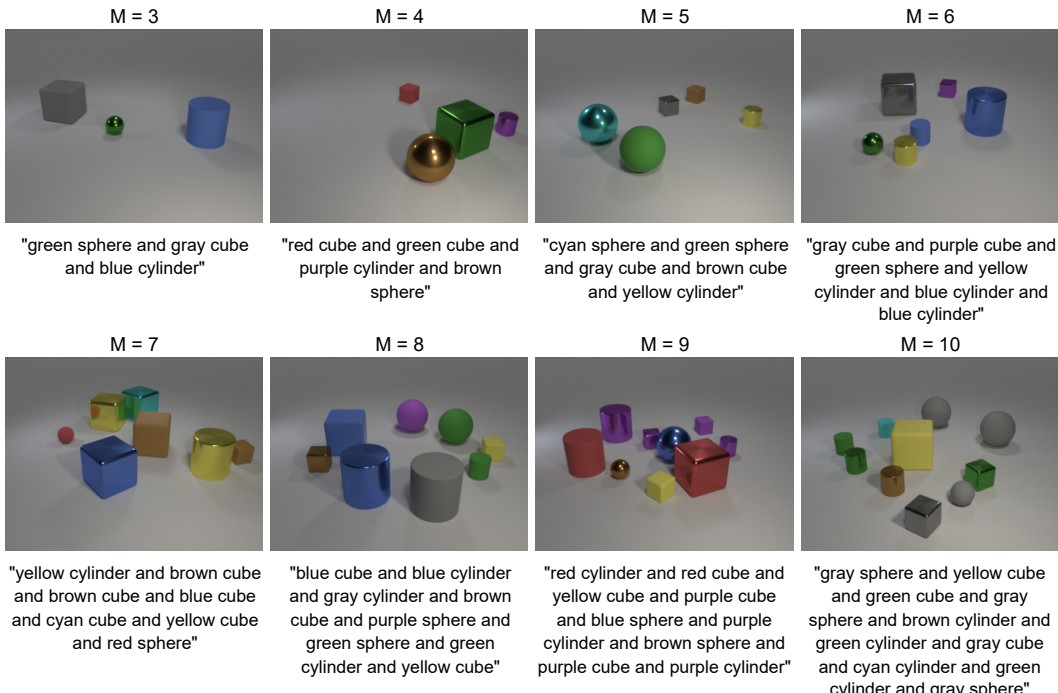

Figure 8: **Scene complexity increases with more objects.** We show examples from the CLEVR with scenes containing different numbers of objects to highlight how image and text complexity changes with object count.

### A.1.2 PUG:SPAR

**Description**. PUG:SPAR is a synthetically generated dataset in Unreal Engine Bordes et al. (2024), featuring animal figures on various backgrounds. The animals can have their natural colors or attributes such as red, blue, grass, and stone. For our project, which tests attribute-object relationships in a controlled environment, we filter the dataset to include scenes with two animals and annotated attributes. This leaves us with images of two objects either in a blue/red or grass/stone attribute setting, with one animal on the left and the other on the right. However, the attributes are fixed to positions: the left objects are always blue or grass, and the right objects are always red or stone. This results in $32 \times 31 \times 2 = 1984$ attribute-object combinations and a total of $19840$ images.

As discussed in Section 3.2, the fixed relationship between attributes and positions could lead to a shortcut strategy for the linear classifier: first identifying the blue/red or grass/stone setting, and then determining if the target object is on the left or right. To address this, we create PUG:SPARE, a dataset with an extended set of attributes that are independent of object positions.

**Train/test split**. Similar to the CLEVR dataset, we split the dataset into training, validations, and test sets based on attribute-object combinations in a 90/10/10 ratio.

### A.1.3 PUG:SPARE

**Generation**. Similar to PUG:SPAR, we generate photorealistic images of two animals on different backgrounds. Our dataset includes 12 possible animal objects and 8 possible colors for these animals. The objects appear in 4 different environments, creating varied backgrounds and lighting conditions. The relative positions between the two objects also change: the left object in front and the right object in the back, the left object in the back and the right object in front, and both objects at the same distance. Animals and attributes do not repeat. For example, "red zebra and blue lion" is valid. However, "red zebra and red lion" or "red zebra and blue zebra" are not. We generate all possible two-object combinations with these conditions. This results in 12 choices for the first object, 11 choices for the second object, 8 colors for the first object, 7 colors for the second object, divided by 2 to ignore the order: $12 \times 11 \times 8 \times 7 \times 0.5 = 3696$ combinations.

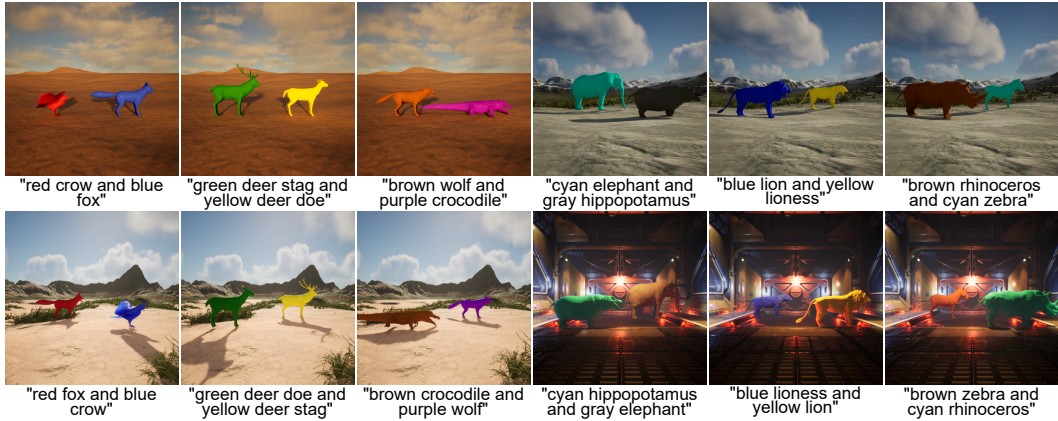

Figure 9: **PUG:SPARE dataset examples.** PUG:SPARE offers all possible configurations of two objects from the set of 12 objects, 8 attributes, 4 backgrounds, and 3 position configurations (front/back, back/front, the same distance). This allows comprehensive testing of attribute-object binding.

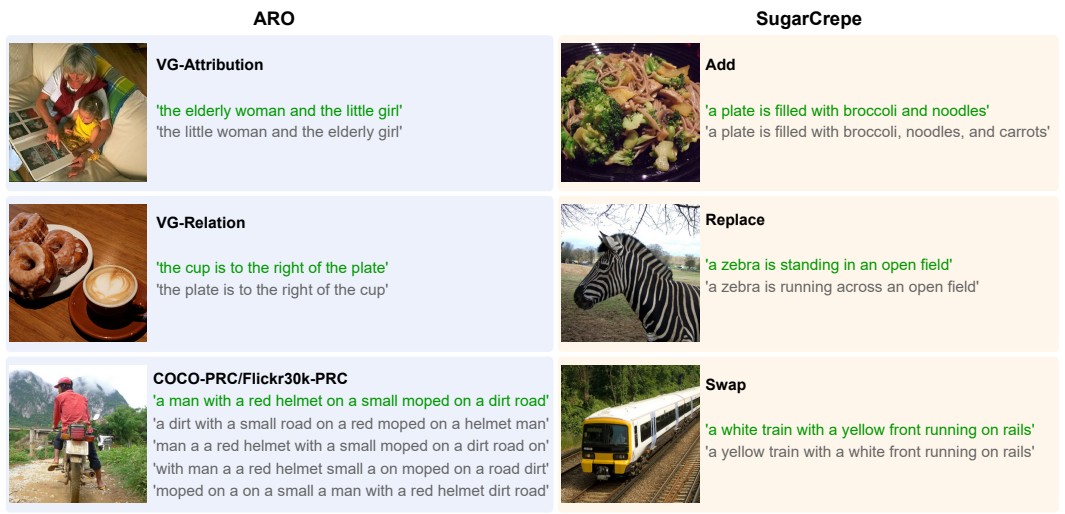

Figure 10: **Examples from the compositional benchmarks ARO and SugarCrepe.** These datasets are designed to test VLMs' ability to accurately bind attributes and objects in complex, real-world scenarios. The images illustrate varied compositions of objects, attributes, and their relationships, challenging a model's compositional understanding.

The images for these combinations are rendered in Unreal Engine. The image dimensions are $512 \times 512$. Captions follow the form "$a_1$ $o_1$ *and* $a_2$ $o_2$", where $a_j \in \mathcal{A}$ and $o_j \in \mathcal{O}$. The examples are shown in Fig. 9.

**Train/test split**. Similar to the CLEVR dataset, we divide the dataset into train, validation, and test splits based on the attribute-object combinations.

## A.1.4   COCO, ARO, SUGARCREPE

We evaluate the cross-modal binding performance of our method on real-world datasets by training on COCO Lin et al. (2014). Following the protocol in Yuksekgonul et al. (2023), we apply the Karpathy splits to divide the dataset into train, validation, and test sets.

The trained models are then assessed on compositional benchmarks, ARO Yuksekgonul et al. (2023) and SugarCrepe Hsieh et al. (2024). Examples from these benchmarks are shown in Fig. 10.

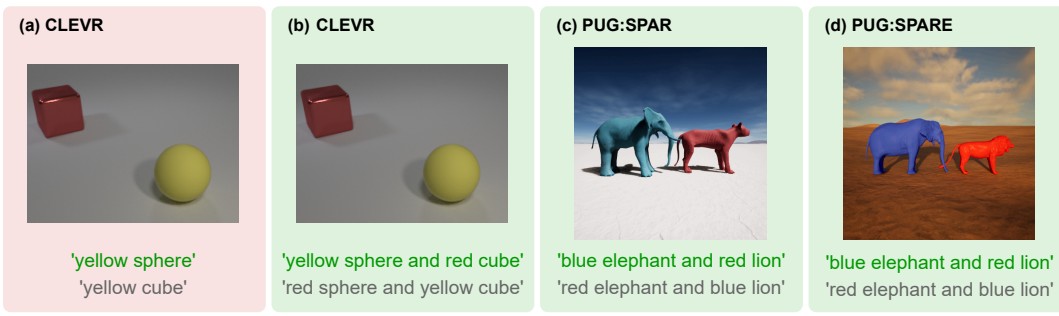

Figure 11: **Illustrating BoWness.** The figure presents examples from CLEVR, PUG:SPAR, and PUG:SPARE datasets, used to demonstrate BoWness. The highlighted example in red shows a case where an image is compared to captions describing only a single object, which can lead to inaccurate assessments due to incomplete scene information.

## A.2  DISCUSSION ON ESTABLISHING BOWNESS

Previous studies evaluating CLIP's Bag-of-Words (BoW) behavior have typically combined both text and image modalities, following the standard zero-shot learning approach. These assessments generally fall into two categories. The first approach, as demonstrated in Lewis et al. (2024); Tang et al. (2023), involves comparing images to captions that describe only a single object in the scene. For example, given an image with a yellow sphere and a red cube, the text prompts might be: 'a photo of {yellow sphere, yellow cube, red sphere, blue cube, purple cylinder}' (Fig. 11(a)). We argue that focusing on single-object descriptions does not fully capture CLIP's capability for compositional reasoning in multi-object contexts. Since CLIP is trained to match an image to the caption that best aligns with its content, 'yellow cube' would be a better match than 'yellow sphere' in this example, as it provides a more accurate representation of the concepts in the scene. Limiting the evaluation to single-object descriptions, therefore, does not fairly test its ability to understand attribute-object associations.

The second approach, seen in studies like Yuksekgonul et al. (2023), provides a more robust evaluation by comparing a query image to permutations of a more complete description, effectively testing BoWness. We adopt this methodology for our experiments in Section 3.3. Specifically, we measure CLIP's accuracy in choosing between correct and permuted captions. For CLEVR, if the correct caption is 'yellow sphere and red cube', we compare it to its permutation, 'red sphere and yellow cube'(Fig. 11(b)). Similarly, for the PUG:SPAR and PUG:SPARE datasets, we test the model's ability to distinguish between captions like 'blue elephant and red lion' versus 'red elephant and blue lion' (Fig. 11(c),(d)).

## A.3  DETAILS FOR UNI-MODAL BINDING

In this section, we provide details about the experiments conducted in Section 4.

Since each linear probe is object-specific, we filter the dataset to include only examples containing the target object. These examples are split into training, validation, and test sets with a 90/10/10 ratio based on the attribute-object combinations that include the target object.

We use OpenAI's CLIP model for all experiments Radford et al. (2021). The main results presented in Table 2 are based on the ViT-L/14 model, while additional results with the ViT-B/32 and ViT-B/16 models are shown in Table 6. During linear probing, the model weights are frozen. For upper-bound results derived with fine-tuning, we unfreeze the

Table 6: **CLIP is not BoW uni-modally.** The table shows accuracies of linear probes classifying attributes for each target object, averaged across all objects. Results extend Table 2 with ViT-B/32 and ViT-B/16 backbones.

| Dataset | Encoder | Image | | Text | |
|---|---|---|---|---|---|
| | | Train | Test | Train | Test |
| CLEVR | ViT-B/32 | 1.0 | 0.91 | 1.0 | 0.99 |
| | ViT-B/16 | 1.0 | 0.90 | 1.0 | 0.99 |
| PUG:SPAR | ViT-B/32 | 1.0 | 0.97 | 1.0 | 0.98 |
| | ViT-B/16 | 1.0 | 0.98 | 1.0 | 0.99 |
| PUG:SPARE | ViT-B/32 | 0.96 | 0.90 | 1.0 | 0.99 |
| | ViT-B/16 | 0.96 | 0.92 | 1.0 | 0.99 |

Table 7: **The results of the LABCLIP-SB on real-world benchmarks.** LABCLIP-SB does not contain hard negative samples in batches. Its performance is lower than LABCLIP-HNB on ARO and similar on average for SugarCrepe and COCO retrieval.

| | | ARO | | | | SugarCrepe | | | COCO |
|---|---|---|---|---|---|---|---|---|---|
| Model | Backbone | VG-A | VG-R | Flickr PRC | COCO PRC | Add | Replace | Swap | Recall@1 |
| LABCLIP-SB | ViT-B/32 | 0.64 | 0.59 | 0.42 | 0.32 | 0.83 | 0.83 | 0.69 | 0.41 |
| LABCLIP-SB | ViT-B/16 | 0.60 | 0.57 | 0.41 | 0.32 | 0.84 | 0.84 | 0.67 | 0.44 |
| LABCLIP-SB | ViT-L/14 | 0.62 | 0.60 | 0.44 | 0.31 | 0.85 | 0.84 | 0.64 | 0.46 |

relevant image or text encoder weights and allow them to update during training. All models are trained on a single Nvidia A100 GPU.

We do not normalize image or text embeddings before passing them to the linear classifiers. The linear classifiers are implemented in PyTorch using cross-entropy loss. Accuracy is measured by predicting attributes and averaging the results across all object-specific classifiers. In the two-object case, each object has only one possible color, and the task is to predict a single color per object. In the multi-object case, where multiple instances of the target object may appear, a prediction is considered correct only if all colors for the given target object are accurate.

We use both manual and random searches for hyperparameter tuning. A batch size of 32 consistently performs well across all datasets. For CLEVR, linear probing without fine-tuning requires learning rates of $\{0.1, 0.01, 0.001\}$ and training for 1000 to 5000 epochs for images or 200 to 1000 epochs for text. For fine-tuning, we reduce the number of epochs to a range of 5 to 20. For the PUG datasets, we maintain a batch size of 32, with a learning rate of 0.1, and train for 50 to 200 epochs when not fine-tuning. For fine-tuning, the learning rates range between 0.001 and 0.01, with 5 to 50 epochs. Both training regimes utilize the SGD optimizer. The optimal configurations are selected based on validation set accuracy.

**Details on training a BoW model.** We simulate a BoW model by training CLIP encoders to recognize all attributes in the input while ignoring binding to objects.

We attach a linear layer to CLIP's image or text encoder that maps to attribute classes, similar to linear probes. We then reinitialize the CLIP encoders randomly and train the model to predict all attributes in the input with soft label cross-entropy loss. The soft labels correspond to the normalized count of attributes in the input. This ensures that the model behaves as a BoW because it is tasked to predict attributes without having to link them to specific objects. We use these newly trained CLIP embeddings and apply linear probing to evaluate the presence of attribute-object binding information.

As explained in the main text, such a BoW model does not achieve high linear probing accuracy. On CLEVR, the average test accuracies of the linear probes are 0.66 for images and 0.85 for text, significantly worse than the probing performance on the actual CLIP embeddings (0.96). This reinforces the idea that BoW models do not contain features that are useful for binding.

### A.4 CONJUNCTIVE SEARCH DETAILS

We trained separate linear classifiers on CLIP's visual embeddings for various object cases (5, 10, 15, ..., 50 objects). The visual embeddings were extracted using the ViT-L/14 encoder. The classifiers predicted whether an image contained a red sphere (or a red "L" in the letters setting). Training was performed on 800 images per case, with evaluation on a test set of 100 images. In addition, we conducted a zero-shot classification with captions. For the incongruent case, the captions were of the form "This image contains *a red sphere* and green spheres and red cubes". For the congruent case, the caption was "This image contains green spheres and red cubes".

### A.5 CROSS-MODAL BINDING DETAILS

In this section, we provide additional details about LABCLIP, discussed in Section 5. The training process of LABCLIP is illustrated in Algorithm 1.

---

**Algorithm 1** Training algorithm for Linear Attribute Binding CLIP (LABCLIP)

---

1: Initialize transformation matrix $\mathbf{A} \in \mathbb{R}^{D \times D}$
2: **Precompute** $\mathbf{i}_i = f_{\text{image}}(\mathbf{x}_i^{\text{img}})$ and $\mathbf{t}_i = f_{\text{text}}(\mathbf{x}_i^{\text{txt}})$ for all $i$ (CLIP encoders frozen)
3: **for** epoch = 1 to $N_{\text{epochs}}$ **do**
4:     **for** each batch $\{(\mathbf{i}_i, \mathbf{t}_i)\}_{i=1}^B$ **do**
5:         $\mathbf{t}_{i,\text{pos}} = \mathbf{A}\mathbf{t}_i$
6:         Compute positive scores $s_{i,i} = \langle \mathbf{i}_i, \mathbf{t}_{i,\text{pos}} \rangle$
7:         **if** negative sampling **then**
8:             Generate negatives: $\mathbf{t}_{j,\text{neg}} = \mathbf{A}\, f_{\text{text}}(\text{permute}(\mathbf{x}_j^{\text{txt}}))$
9:             Compute negative scores $s_{i,j} = \langle \mathbf{i}_i, \mathbf{t}_{j,\text{neg}} \rangle$ for $i \neq j$
10:           **Effective batch size**: $B \times 2B$
11:         Compute $\mathcal{L}_{\text{img-to-txt}} = \text{CE}(\{s_{i,i}\}, \{s_{i,j}\})$
12:         Compute $\mathcal{L}_{\text{txt-to-img}} = \text{CE}(\{s_{i,i}\}, \{s_{j,i}\})$
13:         $\mathcal{L} = \mathcal{L}_{\text{img-to-txt}} + \mathcal{L}_{\text{txt-to-img}}$
14:         Update $\mathbf{A}$ to minimize $\mathcal{L}$
15: **Return** learned transformation matrix $\mathbf{A}$

---

Table 8: **LABCLIP enhances cross-modal binding.** These results extend the findings shown in Table 3 for ViT-B/32 and ViT-B/16 backbones.

| Model | Backbone | Accuracy | | Recall@1 | |
|---|---|---|---|---|---|
| | | Train | Test | Train | Test |
| **CLEVR** | | | | | |
| CLIP | ViT-B/32 | 0.52 | 0.49 | 0.33 | 0.34 |
| LABCLIP-SB | ViT-B/32 | 0.99 | 0.85 | 0.99 | 0.83 |
| LABCLIP-HNB | ViT-B/32 | 0.99 | 0.83 | 0.99 | 0.81 |
| CLIP | ViT-B/16 | 0.50 | 0.54 | 0.31 | 0.38 |
| LABCLIP-SB | ViT-B/16 | 1.00 | 0.93 | 1.00 | 0.92 |
| LABCLIP-HNB | ViT-B/16 | 1.00 | 0.93 | 1.00 | 0.92 |
| CLIP | ViT-L/14 | 0.49 | 0.58 | 0.25 | 0.36 |
| LABCLIP-SB | ViT-L/14 | 1.00 | 0.95 | 1.00 | 0.94 |
| LABCLIP-HNB | ViT-L/14 | 1.00 | 0.95 | 1.00 | 0.93 |
| **PUG:SPAR** | | | | | |
| CLIP | ViT-B/32 | 0.51 | 0.51 | 0.02 | 0.02 |
| LABCLIP-SB | ViT-B/32 | 0.99 | 0.97 | 0.93 | 0.83 |
| LABCLIP-HNB | ViT-B/32 | 0.99 | 0.98 | 0.93 | 0.84 |
| CLIP | ViT-B/16 | 0.52 | 0.53 | 0.04 | 0.04 |
| LABCLIP-SB | ViT-B/16 | 0.99 | 0.97 | 0.94 | 0.88 |
| LABCLIP-HNB | ViT-B/16 | 1.00 | 0.98 | 0.95 | 0.89 |
| CLIP | ViT-L/14 | 0.52 | 0.53 | 0.08 | 0.09 |
| LABCLIP-SB | ViT-L/14 | 1.00 | 0.97 | 0.98 | 0.90 |
| LABCLIP-HNB | ViT-L/14 | 1.00 | 0.97 | 0.98 | 0.91 |
| **PUG:SPARE** | | | | | |
| CLIP | ViT-B/32 | 0.51 | 0.51 | 0.01 | 0.01 |
| LABCLIP-SB | ViT-B/32 | 0.91 | 0.89 | 0.73 | 0.69 |
| LABCLIP-HNB | ViT-B/32 | 0.95 | 0.93 | 0.77 | 0.73 |
| CLIP | ViT-B/16 | 0.51 | 0.50 | 0.03 | 0.03 |
| LABCLIP-SB | ViT-B/16 | 0.91 | 0.87 | 0.84 | 0.80 |
| LABCLIP-HNB | ViT-B/16 | 0.96 | 0.92 | 0.88 | 0.84 |
| CLIP | ViT-L/14 | 0.50 | 0.50 | 0.06 | 0.06 |
| LABCLIP-SB | ViT-L/14 | 0.94 | 0.90 | 0.90 | 0.86 |
| LABCLIP-HNB | ViT-L/14 | 0.98 | 0.94 | 0.95 | 0.90 |

We use OpenAI's CLIP for all experiments, specifically the L/14 model for CLEVR, PUG:SPAR, and PUG:SPARE, as shown in Table 3. Additional results for the B/32 and B/16 models are in Table 8. All models were trained on a single Nvidia A100 GPU.

The LABCLIP method involves adding an additional linear layer to the text encoder while keeping the CLIP weights frozen. For the upper bound results with fine-tuned CLIP, we unfreeze both encoder weights. The additional linear layer has the same dimension as the network output, equivalent

to multiplying by a $D \times D$ matrix. We initialize this matrix as an identity matrix, which corresponds to the case when no transformation is applied. The matrix is trained without further constraints.

Table 9: **Aligning image space instead of text space in LABCLIP yields similar results.** Test accuracies across all datasets. Performance improves for synthetic datasets and remains similar for real-world datasets, confirming invariance to the projection's location.

| Dataset | Accuracy |
|---|---|
| CLEVR | 0.98 |
| PUG:SPAR | 0.99 |
| PUG:SPARE | 0.96 |
| VG-A | 0.68 |
| VG-R | 0.82 |
| Flickr-PRC | 0.84 |
| COCO-PRC | 0.81 |
| Add | 0.82 |
| Replace | 0.82 |
| Swap | 0.72 |
| COCO R@1 | 0.42 |

We use the same contrastive loss as in the original CLIP. The temperature parameter in the contrastive loss is a learnable parameter, initialized to 0. Two approaches are explored: the **Standard Batch (SB)** contains only the corresponding images and text in each batch, while the **Hard Negative Batch (HNB)** also includes hard negative captions. We use the approach with Hard Negative Batch as the default version of LABCLIP. We report results for LABCLIP-SB in Tables 8 and 7.

For synthetic datasets, we obtain hard negatives by swapping attributes. For COCO, we utilize the attribute and noun shuffling method introduced in NegCLIP to create negative captions Yuksekgonul et al. (2023). For example, in COCO-PRC example shown in Fig. 10, when the correct caption is 'a man with a red helmet on a small moped on a dirt road', a possible negative caption is 'a dirt with a small road on a red moped on a helmet man'. There are no negative images in our approach.

We employ a combination of manual and random searches for hyperparameter tuning, with the batch size ranging from 32 to 2048, epochs from 5 to 50, and learning rates between 0.0001 and 0.01. We use the Adam optimizer for LABCLIP but SGD when fine-tuning on the CLEVR and PUG datasets. The optimal hyperparameters are selected based on the final epoch performance on the validation set.

**Aligning the image space.** We chose to apply the alignment matrix to the text embeddings. However, applying the projection to the text or image embeddings is conceptually equivalent. Let $u, v \in \mathbb{R}^D$ be image and text embeddings. Our learned matrix $A \in \mathbb{R}^{D \times D}$ is applied on $v$ before inner product: $sim = u^\top (Av)$. Note the equivalence: $sim = (A^\top u)^\top v$, which signifies the application of the linear matrix $A^\top$ on the image embeddings. Additional experiments with aligning the image space confirms the invariance of the results to where the projection is applied (See Table 9). The results improve for synthetic datasets and remain similar for real-world datasets.

**Comparison to CLEVR by Lewis et al. (2024)**. Our two-object CLEVR dataset was inspired by the two-object setup in (Lewis et al., 2024) as it allows for clear control of attributes and objects. However, the main difference between the works is the division of compositions into train, validation, and test splits. In (Lewis et al., 2024), some attribute-object pairs (for example, brown cube) are held out, so no scene containing a brown cube appears in training. In contrast, we split by two-object combinations: "brown cube and blue sphere" may appear in training, while "brown cube and red cylinder" does not. Holding out attribute-object pairs reduces training diversity, and prior work (Uselis et al., 2025) shows that such diversity is important for compositional generalization.

Training LABCLIP on the dataset from (Lewis et al., 2024) yields similar results (our LABCLIP 1.29% vs their fine-tuned CLIP 0.25%), showing the difficulty of compositional generalization. To isolate binding, we shuffle the splits so every attribute-object pair appears in training. The evaluation still tests binding across scenes because position, material, and orientation vary. LABCLIP's performance, therefore, reflects whether CLIP's embeddings contain a reliable attribute-object signal that holds across these variations. In this setting, LABCLIP reaches 94.3% on the two-object dataset, indicating that the embeddings contain a strong, recoverable binding signal.

**Spatial reasoning.** We evaluate spatial reasoning to test whether improved attribute-object binding transfers to spatial reasoning. Spatial relations describe how objects relate to each other or to the scene, while attribute-object binding concerns how an object is linked to its attributes. These are distinct challenges. Spatial rea-

Table 10: **LABCLIP trained only on binding matches CLIP's spatial reasoning and improves on spatial tasks when given spatial supervision.** We evaluate LABCLIP trained on COCO and CC3M with attribute-object hard negatives, and LABCLIP trained with spatial hard negatives on half of each dataset, on the What'sUp.

| Model | Training data | What'sUp | COCO-Spatial | GQA-Spatial |
|---|---|---|---|---|
| CLIP | – | 0.31 | 0.47 | 0.47 |
| LABCLIP | COCO | 0.31 | 0.48 | 0.46 |
| LABCLIP | CC3M | 0.31 | 0.51 | 0.45 |
| LABCLIP | Spatial data | 0.54 | 0.64 | 0.55 |

soning is difficult for VLMs because spatial data is rare in training corpora (Kamath et al., 2023b)

and spatial descriptions can be ambiguous (Liu et al., 2023a), which makes cross-modal alignment harder.

We evaluate LABCLIP on the What'sUp benchmark (Kamath et al., 2023b) in Table 10. LABCLIP trained only on attribute-object binding achieves performance comparable to CLIP (LABCLIP-CC3M 0.31 vs CLIP 0.31 on What'sUp, 0.51 vs 0.47 on COCO-Spatial, 0.45 vs 0.47 on GQA-Spatial), showing that it matches CLIP's level of spatial reasoning. To test whether the method can extract spatial relations with supervision, we train LABCLIP on half of each dataset and evaluate on the remaining half. LABCLIP-spatial improves across all datasets (0.54, 0.64, 0.55), although the scores remain below human-level accuracy, consistent with Kamath et al. (2023b).

**Performance on downstream tasks.** We show the zero-shot and probing performance for downstream tasks before and after LABCLIP in Table 11.

LABCLIP, both when trained on COCO and CC3M, shows lower accuracy on CIFAR10, CIFAR100, and ImageNet in the zero-shot setting than CLIP. This is expected, as LABCLIP is trained with hard negative captions that shuffle nouns and adjectives, making it more effective for compositional retrieval tasks. In contrast, CIFAR10, CIFAR100, and ImageNet rely on single-object captions, which LABCLIP is not specifically optimized for. The stronger performance of NegCLIP in these cases may stem from its use of hard nega-

Table 11: **LABCLIP preserves probing performance but reduces zero-shot accuracy on single-object datasets.** Since LABCLIP does not modify image embeddings, probing results remain unchanged.

|  | CLIP | NegCLIP | LABCLIP (COCO) | LABCLIP (CC3M) |
|---|---|---|---|---|
| **Zero-shot** | | | | |
| CIFAR 10 | 0.90 | 0.89 | 0.86 | 0.89 |
| CIFAR 100 | 0.65 | 0.63 | 0.57 | 0.63 |
| ImageNet | 0.56 | 0.53 | 0.46 | 0.51 |
| Flickr30k T2I | 0.59 | 0.71 | 0.65 | 0.61 |
| Flickr30k I2T | 0.78 | 0.85 | 0.78 | 0.72 |
| COCO T2I | 0.30 | 0.45 | 0.41 | 0.34 |
| COCO I2T | 0.50 | 0.59 | 0.55 | 0.46 |
| **Probing** | | | | |
| CIFAR 10 | 0.95 | 0.94 | 0.95 | 0.95 |
| CIFAR 100 | 0.80 | 0.79 | 0.80 | 0.80 |
| ImageNet | 0.75 | 0.72 | 0.75 | 0.75 |

tive images or fine-tuning. Note that LABCLIP trained on COCO outperforms LABCLIP trained on CC3M on the retrieval tasks. This is expected because dataset overlap between training and testing can boost retrieval performance (Oh et al., 2024). Since LABCLIP does not modify CLIP's original image representations, probing results remain unchanged.

## A.6 ADDITIONAL ANALYSES

### A.6.1 REPRESENTATIONAL SIMILARITIES AND ALIGNMENT

We analyze cosine similarity distributions between positive and negative pairs before and after applying the alignment matrix $\mathbf{A}$ to observe the impact of our alignment transformation. For a text sequence $\mathbf{x}^{\text{txt}}$, the aligned text representation is $\mathbf{A}f_{\text{text}}(\mathbf{x}^{\text{txt}})$, where $\mathbf{A}$ is the alignment transformation applied to the original CLIP text embedding $f_{\text{text}}(\mathbf{x}^{\text{txt}})$.

**Text-to-text similarity.** First, we consider cosine similarities between positive and negative text representations before alignment, $\langle f_{\text{text}}(\mathbf{x}_i^{\text{txt}}), f_{\text{text}}(\text{permute}(\mathbf{x}_i^{\text{txt}}))\rangle_{i=1}^N$, and after alignment, $\langle \mathbf{A}f_{\text{text}}(\mathbf{x}_i^{\text{txt}}), \mathbf{A}f_{\text{text}}(\text{permute}(\mathbf{x}_i^{\text{txt}}))\rangle_{i=1}^N$.

The distributions are depicted in Fig. 12. We observe that, before alignment, the similarities are higher, indicating that image embeddings may be incorrectly matched with permuted text embeddings. After alignment, positive and negative text pairs become more dissimilar, potentially making it easier to distinguish between permuted text pairs.

**Image-to-text similarity.** We analyze cross-modal similarities by comparing image embeddings to both positive and negative text embeddings (Fig. 12). The solid bars represent similarities between

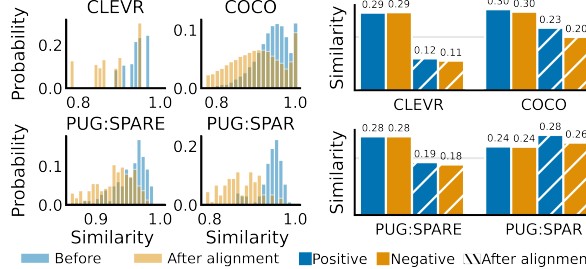

Figure 12: **LABCLIP reduces the similarity between negative pairs.** Left: distributions of cosine similarities between original and permuted captions before and after alignment. Right: mean similarities between positive and negative image–text pairs before and after alignment.

the image and text embeddings for positive pairs $\langle f_{\text{image}}(\mathbf{x}_i^{\text{img}}), f_{\text{text}}(\mathbf{x}_i^{\text{txt}}) \rangle_{i=1}^N$ and negative pairs $\langle f_{\text{image}}(\mathbf{x}_i^{\text{img}}), f_{\text{text}}(\text{permute}(\mathbf{x}_i^{\text{txt}})) \rangle_{i=1}^N$ before alignment.

The results indicate no distinction between positive and negative pairs before alignment, as the solid bars for both are at the same height. However, after alignment (dashed bars), the similarity to positive text embeddings $\langle f_{\text{image}}(\mathbf{x}_i^{\text{img}}), f_{\text{text}}(\mathbf{A}\mathbf{x}_i^{\text{txt}}) \rangle_{i=1}^N$ is notably higher than to permuted text $\langle f_{\text{image}}(\mathbf{x}_i^{\text{img}}), f_{\text{text}}(\mathbf{A}\text{permute}(\mathbf{x}_i^{\text{txt}})) \rangle_{i=1}^N$. This demonstrates that alignment enables better differentiation, allowing the model to match images with the correct text rather than the permuted text.

### A.6.2 IMPLICATIONS TO MODALITY GAP

A key challenge in VLMs like CLIP is the modality gap, a discrepancy between vision and text embeddings Liang et al. (2022). Previous studies Schrodi et al. (2024) suggest that reducing this gap improves interaction between modalities. Motivated by this, we measure the Euclidean distance between mean embeddings $\mathbf{x}$ and $\mathbf{y}$ from the vision and text before and after alignment. Specifically, we define $\mathbf{x} := \frac{1}{N} \sum_{i=1}^N f_{\text{image}}(\mathbf{x}_i)$ and $\mathbf{y} := \frac{1}{N} \sum_{i=1}^N f_{\text{text}}(\mathbf{y}_i)$, where $f_{\text{image}}$ and $f_{\text{text}}$ denote the encoders, and $N$ is the sample size. We then compute $\|\mathbf{x} - \mathbf{y}\|_2$ for original embeddings and $\|\mathbf{x} - \mathbf{A}\mathbf{y}\|_2$ for aligned embeddings, where $\mathbf{A}$ is a matrix trained with LABCLIP.

Our experiments reveal that the modality gap decreases across the COCO, PUG:SPAR, and PUG:SPARE datasets after alignment, while CLEVR shows a slight increase (Fig. 13, left). We also provide a qualitative illustration of the modality gap (Fig. 13, right) using UMAP McInnes et al. (2018) on the COCO test set. In this visualization, the aligned text representations (green) move closer to the image representations (blue), indicating a

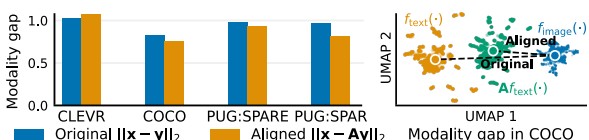

Figure 13: **Modality gap decreases after alignment.** Left: modality gap between image and text representations before and after alignment. Right: UMAP visualization of COCO test set representations with text representations before and after alignment.

reduced modality gap after alignment. These results suggest that alignment effectiveness may vary across datasets, with our approach successfully enhancing cross-modal compatibility in most cases.

### A.7 THE USAGE OF LLMS

In accordance with ICLR 2026 policy, we disclose that large language models were used to assist in text editing and polishing of writing. All research ideas, experiments, and analyses were conducted by the authors.

