# OpenReview forum: "CLIP Behaves like a Bag-of-Words Model Cross-modally but not Uni-modally"
_ICLR.cc/2026/Conference — ICLR 2026 Poster_

### Official Review · Reviewer_CZn4 · 2025-10-26

**Soundness:** 4
**Presentation:** 4
**Contribution:** 4
**Rating:** 8
**Confidence:** 4

**Summary:**

This paper explores the Bag-of-Words aspect of CLIP embeddings, namely that compositional information is limited and representations are not sensitive to compositional orderings in a cross-modal way, but that this information is present in the individual modalities. The work goes on to explore various ideas aimed at uncovering what is at the core of this problem and how to fix the BoW-nature of standard CLIP embeddings in cross-modal compositional contexts.

In summary, I loved reading this paper and the simple explanation and guided tour of the hypothesis, easy-to-understand experiments that are nice and conceptually clear. I definitely think this is highly relevant to the community. However, passages such as the one in the conclusion ("We investigated the reasons behind CLIP's bag-of-words behavior") sort of ring a little bit hollow because I feel like a solid step was taken towards localising the nature of the issue, but I am not so sure a lot of solid ground was covered to explain the more interesting question - why this occurs. We know that a linear transformation among the spaces to align the embeddings goes some way to solving this issue, but what drives this and how simple linear transformations actually "recover compositional information" from a BoW kind of treatment is still a bit of a mystery to me. Having said that, given the space allowed for submissions, this is a great submission and full of detail and interesting ideas.

**Strengths:**

This is quite a well-written paper. It's informative and also understands the audience needs some examples, so the example with the orange square and blue triangle to highlight the problem in a clear example is very welcome to give a concrete example of the issue at hand (and how the permuted text also has a high match with the inconsistent image). The questions that lie at the crux of the issue are put forward in a simple, clear and easy-to-understand manner. This is refreshing to see and greatly appreciated, making it easy to follow how the setup of the experiments answers the questions that truly get to the heart of what the paper claims to be about.

There is clear attention to detail and awareness of the often-unavoidable quagmire of confounds that render any experimental results across a large dataset of real images to be impossible to declare to be fully associated with a list of simple claims. This then pushed the authors to work in a controlled setting that they are able to manipulate clearly and cleverly in order to show that conclusions and differences only arise due to the re-ordering of the components in the text encoder. This experimental awareness is a solid positive for this paper, also in the recognition that both real and synthetic datasets can provide complementary results and a holistic approach of the results is often necessary.

Another strength is the consideration and attention to splits in the linear classifier case. This work correctly identifies previous approaches being problematic and not evaluating where the lack of alignment comes from. Additionally, the presence of summary takeaway points greatly enables the reader to follow the main points being raised and to follow the authors' claims and hypotheses with regard to the presented results.

**Weaknesses:**

The weaknesses here are only very minor and should be treated as small tips to increase the readability or accessibility of the paper.

1. When listing the compositional datasets, it would be helpful to include the names of those benchmarks rather than just the generic theme and the citation, as not everyone will be aware of the citation but might be familiar with a name of a specific dataset and this would only increase the mental links you are raising in those lines of the paper. I know later in Table 1 this is clarified, but you might facilitate reading by including them in that section, too.

2. When discussing the matching of permuted text encoders, it means something different to be picking the correct match out of a small number of possibilities than it is to picking out the correct match among a large potential batch of potential pairs, due to the fact that there is a higher chance to randomly select the correct one. I found myself curious of this when reading through section 3 and felt like additional information would be welcome in this part of the paper, even if just in brackets it mentions how many options were being selected from in the batch. This would then stop me from wondering if the batches were small enough that random correct selection might be a possibility to explain some results.

3. The paper's goals aimed to find the cause of the BoW phenomenon and did a great job of locating it, but the abstract and conclusion lead one to believe that more of a causative finding has been presented. I am impressed by the paper and the results found, but after finishing the paper, I'm not sure I've been enlightened too much as to what is causing this phenomenon. The authors have certainly made a scientific advance by locating where the issue lies, but without tying it to other relevant concepts like the modality gap (which the authors show increases in some cases in one of the appendices for a dataset) then I'm left even more confused as to what brought about the cross-modal lack of alignment (I had previously suspected it was more tightly connected to and explained by the modality gap issue). Additionally, I feel some of that work and potential explanatory notes should be made in the relevant literature section.

**Questions:**

1. Have you considered the cases that extend beyond image-text modalities?

2. What potential pitfalls did you consider in using a synthetically generated benchmark?

3. Why do you think PUG:SPARE in CLIP (ft) achieved higher than PUG:SPAR in the same case (i.e. 1.00 > 0.98) when the PUG:SPARE model was supposed to be more difficult. This is not a hugely important question for the validity of the experiments, but I did find myself having this question when considering the results and I wasn't confident in my hypothesis. Perhaps the authors can share some of their thoughts on this.

---

> ### Author Response · Authors · 2025-11-21
>
> We appreciate the reviewer’s positive assessment of our work. We address the minor concerns below.
>
> **1. Naming benchmarks next to citations will improve readability**
>
> Thank you for the suggestion.  We will update the related works section so that the benchmark names (e.g., ARO, SugarCrepe) appear alongside their citations.
>
> **2. Indicating the number of possible captions for retrieval**
>
> Thank you for pointing this out. In Section 3.3 we report the accuracy of choosing the correct caption out of two options: one with the correct attribute-object pairing, one that has the attributes swapped between objects, as we show in the example with the orange square and the blue triangle. The random baseline is therefore 0.5, and CLIP is close to this baseline for our datasets. We will clarify this in the paper.
>
> **3. What is causing BoWness?**
>
> Thank you for raising this. We believe the cross-modal BoWness comes from how the contrastive objective interacts with the training captions. Captions rarely specify both objects and their attributes. The loss rewards matches based on shared object mentions and does not check whether attributes pair with the correct objects. This view is consistent with object bias reported in prior work (Schrodi et al., 2025). Because the model rarely sees supervision that distinguishes correct from incorrect bindings, it has little incentive to align attribute-object pairs across modalities.
>
> Regarding the modality gap, as you noted, it decreases for three out of four datasets. Prior work also suggests that it is not clear whether reducing the gap is always desirable, since changes in the gap can affect fairness and downstream behavior (Liang et al., 2022). Therefore, we do not consider the modality gap as the cause of the semantic misalignment and the BoW phenomenon.
>
> **Answers to the questions:**
>
> **1. "Have you considered the cases that extend beyond image-text modalities?"**
>
> We focus on image-text because this is where the binding problem is most discussed. However, our analysis is not specific to images or text. It requires that a modality has identifiable objects and attributes and that counterfactual examples can be constructed. In principle, the same probing and alignment procedure could extend to audio-text or video-text models if suitable datasets are available.
>
> **2. "What potential pitfalls did you consider in using a synthetically generated benchmark?"**
>
> We considered several pitfalls when using synthetic data. CLEVR has only basic shapes on neutral backgrounds, which may fall outside the distribution of real images. PUG:SPAR addresses this by using more complex shapes and richer backgrounds, but it still contains correlations between object positions and attributes. These correlations can create shortcuts that do not reflect true binding. For this reason, we constructed PUG:SPARE. It removes positional bias and generates all intersections of background, position, attribute, and shape so that no shortcut can solve the task. We also avoid reusing two-object combinations across splits to prevent memorization. Finally, we check the cross-modal results on real datasets such as ARO and SugarCrepe to confirm our results.
>
> **3. "Why do you think PUG:SPARE in CLIP (ft) achieved higher than PUG:SPAR...?"**
>
> Thank you for the observation. PUG:SPARE includes more attributes, more attribute-object combinations, and greater positional variation than PUG:SPAR. This increased diversity may help the fine-tuned model generalize slightly better, which could explain the difference.
>
>
> **References:**
>
> [a] Schrodi et al. Two effects, one trigger: On the modality gap, object bias, and information imbalance in contrastive vision-language representation learning. ICLR 2025
>
> [b] Liang et al. Mind the Gap: Understanding the Modality Gap in Multi-modal Contrastive Representation Learning. NeurIPS 2022

---

### Official Review · Reviewer_kJCN · 2025-10-31

**Soundness:** 3
**Presentation:** 3
**Contribution:** 1
**Rating:** 4
**Confidence:** 4

**Summary:**

CLIP models often has Bag-of-Words (BoW) property where it treats images and texts as sets of individual concepts and struggles to correctly bind each object and its corresponding attribute pairs. This paper questions the reasons of such BoW property with a series of controlled experiments and proposes a simple and efficient fix to this problem. Specifically, several controlled experiments with linear probing on individual visual and text embeddings from object-attribute binding datasets indicate that CLIP models already encode attribute-object relationships in each modality. In addition, it is shown that such attribute-object concepts are linearly separable on the CLIP's representation space. Based on the results, the problem lies on the cross-modal alignment, and a simple solution is proposed by linearly projecting one embedding during the alignment training. In this way, the performance recovers close to the fully fine-tuned CLIP models while achieving several practical advantages such as computational efficiency and backward compatibility of already-deployed systems.

**Strengths:**

This paper investigates the reason behind the CLIP models' BoW property which is often unexplored from the previous compositional reasoning literature. The controlled experiments and analyses made could be a valuable reference for guiding a better solution for the following works. From the methodological perspective, LABCLIP for appending a linear projection matrix on either image or text embedding is effective to remedy the CLIP models' attribute-object binding capabilities, while it does not need the full fine-tuning of encoders parameters. This can benefit both on the training and inference procedure.

**Weaknesses:**

One major concern lies in the scope of the work with respective to compositional reasoning tasks in several aspects. While the paper mainly focused on a relatively simple settings, attribute-object binding, but it is not yet discussed whether the observations and solutions made in the paper can generalize to a more complex and realistic compositional reasoning benchmarks such as Winoground. In addition, since the oracle method to the proposed solution is NegCLIP, which is trained on both original and hard negative captions, the proposed solution might share the NegCLIP's limitations such as lower performance on challenging compositional reasoning benchmarks such as Winoground and original CLIP's downstream tasks including zero-shot image classification and image-text retrieval. Overall, the method appears to focus on a specific area of compositional reasoning and requires further justifications to a complex and realistic settings.

**Questions:**

Regarding the weaknesses stated above, it is expected to show that the proposed findings and methods can generalize beyond the simple attribute-object binding setting to more complex compositional reasoning tasks, such as Winoground, where the oracle method NegCLIP also show limited performance. In addition, it would be important to show how the proposed LABCLIP method affect the general downstream tasks such as zero-shot image classification and image-text retrieval.

---

> ### Author Response · Authors · 2025-11-21
>
> Thank you for raising these important points.
>
> **"the scope of the work with respect to compositional reasoning tasks"**
>
> We focus on attribute-object binding rather than other compositional tasks. We view it as a fundamental ability that CLIP models still struggle with. Studying all compositional operators at once would mix representation, alignment, and generalization effects, which makes it difficult to isolate the underlying issue.
>
> "**it is not yet discussed whether the observations and solutions made in the paper can generalize to a more complex and realistic compositional reasoning benchmarks such as Winoground**"
>
> Our submission already includes evidence that the method generalizes to real-world compositional benchmarks such as ARO and SugarCrepe (Table 4). The method raises the average score from **0.58** to **0.79** on ARO and from **0.72** to **0.79** on SugarCrepe. The fact that these gains come from a simple linear transformation on text embeddings suggests that real-world embeddings already contain some attribute-object binding information.
>
> **Winoground** contains many cases that do not test attribute-object binding. As observed likewise by [a], Winoground often mixes several types of reasoning, including visually difficult cases, unusual images or text, ambiguous captions, and complex commonsense or world knowledge. We have thus scoped it out in our studies.
>
> We also provide additional results for spatial relations and negation to assess how the method generalizes to other compositional tasks.
>
> **Negation:**
> - We evaluate on the multiple-choice subsets of NegBench [b].
> - LABCLIP, which was trained only on attribute-object binding, does not generalize to negation data. It matches CLIP on COCO MCQ (**39.3** vs **39.3**) and VOC MCQ (**38.7** vs **38.7**).
> - For LABCLIP-neg, we use half of the NegBench subsets for training and the remaining half for testing. With this explicit negation supervision, accuracy increases to **70.1** on COCO MCQ and **81.4** on VOC MCQ.
>
> | Model | COCO MCQ | VOC MCQ |
> | - | - | - |
> | CLIP | 39.3 | 38.7 |
> | LABCLIP | 34.5 | 38.7 |
> | LABCLIP-neg | 70.1 | 81.4 |
>
>
> **Spatial relations:**
> - We evaluate on the What’sUp benchmark [c].
> - LABCLIP, trained only on attribute-object binding, performs similarly to CLIP (**31.4** vs **31.2** on What’sUp; **48.2** vs **47.3** on COCO-Spatial; **45.5** vs **47.3** on GQA-Spatial).
> - LABCLIP-spatial, with explicit spatial supervision (using half of each dataset for training and the rest for testing), improves across all datasets (**53.7**, **64.5**, **54.8**).
>
> | Model           | What'sUp | COCO-Spatial | GQA-Spatial |
> | --------------- | -------- | ------------ | ----------- |
> | CLIP            | 31.2     | 47.3         | 47.3        |
> | LABCLIP         | 31.4     | 48.2         | 45.5        |
> | LABCLIP-spatial | 53.7     | 64.5         | 54.8        |
>
> In summary, LABCLIP, trained only on attribute-object binding, does not generalize to negation or spatial relations. The same linear transformation approach works when given task-specific training data. It would be useful to first study how negation and spatial relations are represented in unimodal embeddings before applying linear or non-linear solutions.
>
> **"how the proposed LABCLIP method affect the general downstream tasks such as zero-shot image classification and image-text retrieval"**
>
> We already provide these results in Table 10 in the Appendix of the original submission.
>
> We ask whether improving attribute-object binding affects performance on downstream tasks, including zero-shot classification on CIFAR10, CIFAR100, and ImageNet, and image-text retrieval on Flickr30k and COCO.
>
> LABCLIP *improves* image-text retrieval but *performs worse* on CIFAR and ImageNet.
>
> The latter two rely on single-object descriptions and test broad object-level discrimination rather than attribute-object binding.
>
> The results suggest a tradeoff between strong object recognition and the ability to bind attributes to objects. One possible explanation is that once we introduce supervision that targets binding, the linear transformation shifts the emphasis from coarse object-level signals to attribute-object binding-related structure. We see this as an interesting direction for future work.
>
> **References:**
>
> [a] Diwan, A., Berry, L., Choi, E., Harwath, D., & Mahowald, K. Why is winoground hard? investigating failures in visuolinguistic compositionality. arXiv preprint arXiv:2211.00768 2022
>
> [b] Alhamoud et al. Vision-language models do not understand negation. CVPR 2025
>
> [c] Kamath et al. What's "up" with vision-language models? Investigating their struggle with spatial reasoning. EMNLP 2024

---

> ### Comment · Reviewer_kJCN · 2025-11-27
>
> Thank you for the detailed comments.
>
> It seems that the models in Table 10 are trained on COCO dataset (with ARO-style hard negative captions). Under this specific setting, the difficulty of certain compositional benchmarks (e.g., ARO and SugarCrepe) and retrieval tasks likely becomes lower.
>
> Specifically, from both training and evaluation side, the image source (e.g., COCO) and the negative caption style are shared, which  can give COCO-trained NegCLIP-like models an advantage over pre-trained CLIP models. For example, prior work [1] reported that image-to-text retrieval performance tends to degrade when models are fine-tuned on datasets other than COCO.
>
> This overlap makes it hard to determine whether the performance gains of LABCLIP come from the improved object-attribute binding ability, or simply benefitting from the similarity in dataset style (both captions and images) between training and evaluation.
>
> At this point, it may be necessary to assess the improvements under a more rigorous experimental setting.
>
> ---
>
> [1] Preserving Multi-Modal Capabilities of Pre-trained VLMs for Improving Vision-Linguistic Compositionality, EMNLP 2024

---

> > ### Author Response · Authors · 2025-12-03
> >
> > Thank you for the constructive feedback.
> >
> > **"overlap makes it hard to determine whether the performance gains of LABCLIP come from the improved object-attribute binding ability, or simply benefitting from the similarity in dataset style between training and evaluation"**
> >
> > - To reduce the overlap between training (COCO) and evaluation (ARO and SugarCrepe), **we trained LABCLIP on CC3M**.
> >
> > - Table R1: LABCLIP trained on CC3M (LABCLIP-CC3M) still _outperforms CLIP in all cases_ $\rightarrow$ **Performance gains come from improved object-attribute binding**, not the dataset similarity.
> >
> > - Table R2, LABCLIP-CC3M shows _improved downstream classification_ performance relative to LABCLIP-COCO. For example, CIFAR-100 increases from 0.57 to 0.63 and ImageNet from 0.46 to 0.51.
> >
> > **Table R1: Updated LABCLIP scores when trained with CC3M dataset.**
> >
> > |         | Training dataset | VG-A | VG-R | Flickr PRC | COCO PRC | Add  | Replace | Swap | COCO R@1 |
> > | ------- | ---------------- | ---- | ---- | ---------- | -------- | ---- | ------- | ---- | -------- |
> > | CLIP    | -                | 0.63 | 0.63 | 0.60       | 0.48     | 0.73 | 0.80    | 0.62 | 0.30     |
> > | LABCLIP | COCO             | 0.69 | 0.82 | 0.84       | 0.81     | 0.81 | 0.82    | 0.74 | 0.41     |
> > | LABCLIP | CC3M             | 0.68 | 0.78 | 0.83       | 0.73     | 0.83 | 0.80    | 0.68 | 0.34     |
> >
> > **Table R2: Updated LABCLIP scores on downstream datasets when trained on CC3M.**
> >
> > |            | CLIP | LABCLIP COCO | LABCLIP CC3M |
> > | ---------- | ---- | ------------ | ------------ |
> > | CIFAR-10   | 0.90 | 0.86         | 0.89         |
> > | CIFAR-100  | 0.65 | 0.57         | 0.63         |
> > | ImageNet   | 0.56 | 0.46         | 0.51         |
> > | Flickr T2I | 0.59 | 0.65         | 0.61         |
> > | Flickr I2T | 0.78 | 0.78         | 0.72         |
> > | COCO T2I   | 0.30 | 0.41         | 0.34         |
> > | COCO I2T   | 0.50 | 0.55         | 0.46         |
> >
> > **"prior work [1] reported that image-to-text retrieval performance tends to degrade when models are fine-tuned on datasets other than COCO"**
> >
> > LABCLIP-COCO performs better on several datasets, in line with the reviewer’s point. LABCLIP-CC3M still performs well, showing that the gains are not tied to COCO alone.
> >
> > **"it may be necessary to assess the improvements under a more rigorous experimental setting."**
> >
> > The CC3M results provide the requested rigorous setting. LABCLIP continues to outperform CLIP without relying on COCO-style overlap, confirming that the gains reflect improved binding rather than dataset similarity. This supports our main finding that CLIP’s embeddings already carry substantial attribute-object binding signals that LABCLIP can realign.
> >
> > We thank the reviewer for the valuable suggestions. §5.2 (Table 4) and Appendix A.5 (Table 11) now include results for LABCLIP trained on CC3M, which makes the evaluation more rigorous and strengthens the paper.
> >
> > ---
> > [1] Preserving Multi-Modal Capabilities of Pre-trained VLMs for Improving Vision-Linguistic Compositionality, EMNLP 2024

---

### Official Review · Reviewer_84zT · 2025-11-01

**Soundness:** 2
**Presentation:** 4
**Contribution:** 2
**Rating:** 4
**Confidence:** 4

**Summary:**

Existing work has questioned CLIP’s ability to bind concepts, suggesting it behaves like a bag-of-words representation. In this paper, the authors find that CLIP can bind attributes to their objects, but only unimodally. They present linear-probing experiments, robustness tests with increasing object counts, and conjunctive-search experiments to show that CLIP’s text and visual representations bind concepts independently. Finally, they train a lightweight layer to patch existing CLIP embeddings and teach the model to bind concepts.

**Strengths:**

The paper is well written. The introduction clearly motivates an important problem and explains how prior work does not fully investigate the source of bag-of-wordness. The related-work section is thorough and effectively positions the paper within the literature.

Large-scale experiments: The paper shows that concept binding can be improved with a simple linear probe on large-scale datasets such as ARO and SugarCrepe.

**Weaknesses:**

**Differences in results in prior work.**
While the paper’s results are encouraging, I would like to understand the differences from existing works that show that CLIP, even after fine-tuning, does not bind concepts [a, b].

In [a], the authors claim that the representations are not expressive enough to bind concepts. They introduce the Concept Binding Benchmark and show that fine-tuning CLIP and linear probes (the type-logical model in their experiments) do not generalize to held-out compositions that require binding. However, the results in this manuscript suggest that binding can be remedied with a linear probe.

Furthermore, the results in this paper contradict those of Kamath et al. (2024) [b]. In Table 2 (Section 4.3), [b] fine-tune CLIP and evaluate performance on held-out compositions for spatial reasoning (which can be viewed as binding objects to their relations).

It would be helpful if the authors could explain why the experiments in this paper contradict prior work. The paper would be strengthened by running evaluations on datasets [a] and [b] and demonstrating improved binding.


**Minor fixes**

Line 309: use \citet for Campbell et al., 2024.

Figure 5: Could you draw a box to highlight the incongruency?

**References**

[a] Does CLIP Bind Concepts? Probing Compositionality in Large Image Models. Findings of EACL 2024.

[b] What's "up" with vision-language models? Investigating their struggle with spatial reasoning. EMNLP 2024.

**Questions:**

Do you suspect that the BOWness arises from the contrastive loss or the dual encoder in CLIP? It would be insightful to the reader if you could include this discussion in the paper.

---

> ### Author Response · Authors · 2025-11-21
>
> Thank you for the thoughtful feedback.
>
> Rather than contrasting with prior work, **our study is motivated by [a] and [b]**. Given the lack of cross-modal binding described in [a, b], we question whether this stems from 1) a lack of binding information in embeddings or 2) the alignment of this information across modalities. We find, for the first time, that 2) is the case. Distinguishing the two is essential for addressing the actual limitations. In practice, missing information in the embeddings may require changes to the encoders or training, while misalignment calls for alignment-focused solutions.
>
> **"In [a], the authors claim that the representations are not expressive enough to bind concepts"**
>
> [a] evaluates binding cross-modally, so representation and alignment effects are mixed. We assess the representations’ expressivity to bind concepts directly and unimodally.
>
> **"They [...] show that fine-tuning CLIP and linear probes do not generalize to held-out compositions that require binding"**
>
> The division of compositions into train/val/test splits is exactly what distinguishes the works. In [a], some attribute-object pairs (for example, brown cube) are held out, so no scene containing a brown cube appears in training. In contrast, we split by two-object combinations: 'brown cube and blue sphere' may be in training, while 'brown cube and red cylinder' is not. Holding out attribute-object pairs reduces training diversity, but prior work [c] shows that such diversity is important for compositional generalization.
>
> - When we train LABCLIP on [a]’s split, our results *do not contradict*: **81.6%** vs their reported **78.5%** on the 1-object test,  **1.29%** vs **0.25%** on 2-object, and **0%** vs **0%** on relational test split.
> - When we train LABCLIP on all combinations (shuffled split), the task disentangles binding from compositional generalization. LABCLIP then recovers high performance across all cases: **99.9%** on 1-object, **94.3%** on 2-object, and **84.9%** on relational test set. This shows that when all attribute-object combinations are available, the model can align binding cross-modally.
>
> | Model                | 1-obj train | 1-obj test | 2-obj train | 2-obj test | Rel train | Rel test |
> | ---------------------- | ----------- | ---------- | ----------- | ---------- | --------- | -------- |
> | CLIP                   | 96.8        | 95.0       | 30.5        | 30.8       | 23.8      | 0        |
> | LABCLIP [a]'s split    | 100         | 81.6       | 98.3        | 1.29       | 93.3      | 0        |
> | LABCLIP shuffled split | 100         | 99.9       | 96.2        | 94.3       | 87.3      | 84.9     |
>
>  **"the results in this paper contradict those of [b]"**
>
> We believe our results _do not contradict_ because they address different questions. [b] studies spatial relations, which describe how objects relate to each other or to the scene. Our work focuses on attribute-object binding, that is, how an object is linked to its attributes. These are distinct challenges. Spatial reasoning is difficult for VLMs because spatial data is rare in training corpora [b] and spatial descriptions can be ambiguous [d], which makes cross-modal alignment even harder.
>
> We still run the evaluations as requested:
> - LABCLIP, trained only on attribute-object binding, gives similar performance to CLIP ( **31.4** vs **31.2** on What’sUp; **48.2** vs **47.3** on COCO-Spatial; **45.5** vs **47.3** on GQA-Spatial). This indicates that better binding does not automatically improve spatial reasoning.
> - To test whether LABCLIP can extract spatial relations when given spatial training, we also train it with spatial supervision using half of each dataset. LABCLIP-spatial improves across all datasets (**53.7**, **64.5**, **54.8**). However, in line with [b], the scores remain below human-level accuracy.
>
> | Model           | What'sUp | COCO-Spatial | GQA-Spatial |
> | --------------- | -------- | ------------ | ----------- |
> | CLIP            | 31.2     | 47.3         | 47.3        |
> | LABCLIP         | 31.4     | 48.2         | 45.5        |
> | LABCLIP-spatial | 53.7     | 64.5         | 54.8        |
>
> **Minor fixes** will be addressed in the updated file.
>
> **"Do you suspect that the BOWness arises from the contrastive loss or the dual encoder in CLIP?"**
>
> Since the unimodal encoders already contain binding information, we suspect that the BoWness comes from the training data and the contrastive loss, not from the dual-encoder design. Captions rarely specify both objects and attributes, so the loss rewards shared object mentions rather than correct bindings.
>
> **References:**
>
> [a] Lewis et al. Does CLIP Bind Concepts? Probing Compositionality in Large Image Models. EACL 2024
>
> [b] Kamath et al. What's "up" with VLMs? Investigating their struggle with spatial reasoning. EMNLP 2024
>
> [c] Uselis et al. Does Data Scaling Lead to Visual Compositional Generalization? ICML 2025
>
> [d] Liu et al. Visual Spatial Reasoning. ACL 2023

---

> > ### Comment · Reviewer_84zT · 2025-11-25
> > **Reply to the authors**
> >
> > Thank you for explaining how your work relates to existing work and the additional experiments. Your response has clarified some of my questions and concerns. For that reason, I will bump up the score.
> >
> > Below, I have listed a few more concerns and suggestions:
> > - Your response regarding how your work differs from [b] is quite insightful. Please consider including the differences in the manuscript, as this will help position your work.
> >
> > - The results on [a]’s dataset are quite useful. You are right that your work builds on the findings of [a]. However, I fully do not agree with the motivation behind the shuffled setup. What does the model test when “all attribute-object combinations are available”? Could you include more details about the shuffled experiment, either in the response or the paper?
> >
> > [a] Lewis et al. Does CLIP Bind Concepts? Probing Compositionality in Large Image Models. EACL 2024
> >
> > [b] Kamath et al. What's "up" with VLMs? Investigating their struggle with spatial reasoning. EMNLP 2024

---

> > > ### Author Response · Authors · 2025-12-03
> > >
> > > Thank you for your time and your feedback.
> > >
> > > **"Your response regarding how your work differs from [b] is quite insightful. Please consider including the differences in the manuscript, as this will help position your work."**
> > >
> > > We appreciate the suggestion. In the revised submission, we added the comparison to [a] and [b] in Appendix A.5, with references at Lines 430 and 461.
> > >
> > > **"Could you include more details about the shuffled experiment"**
> > >
> > > In the shuffled experiment, we take all samples, shuffle them, and form new train, validation, and test splits. This ensures that all attribute-object combinations (e.g., brown cube) are seen during the training.
> > >
> > > **"What does the model test when “all attribute-object combinations are available”?"**
> > >
> > > - When every attribute-object pair appears in training, we still evaluate binding **across scenes**. The scenes vary in position, material, and orientation for all attribute-object pairs.
> > > - LABCLIP’s results then reflect whether the embeddings contain a **reliable signal** linking each object to its attribute and whether this signal holds across scene changes. It does not test whether the model can align binding for new attribute-object combinations.
> > > - In [a]’s split, some pairs never appear in training, so errors come from failures in compositional generalization (to new attribute-object combinations) rather than binding.
> > >
> > > We hope this clarification addresses the concern about the motivation behind the shuffled setup.
> > >
> > > ---
> > >
> > >
> > > [a] Lewis et al. Does CLIP Bind Concepts? Probing Compositionality in Large Image Models. EACL 2024
> > >
> > > [b] Kamath et al. What's "up" with VLMs? Investigating their struggle with spatial reasoning. EMNLP 2024

---

### Official Review · Reviewer_dUnx · 2025-11-01

**Soundness:** 2
**Presentation:** 2
**Contribution:** 2
**Rating:** 4
**Confidence:** 3

**Summary:**

This paper investigates the root cause of CLIP's well-documented "bag-of-words" (BoW) behavior in compositional reasoning tasks, where it fails to correctly bind attributes to their corresponding objects in multi-object scenes. The authors make a key distinction between uni-modal and cross-modal binding. Through a series of linear probing, robustness, and conjunctive search experiments, they convincingly demonstrate that CLIP's individual image and text encoders do encode attribute-object binding information in a linearly separable manner. The core problem, they argue, lies not in a lack of information but in a cross-modal alignment failure that prevents this information from being effectively utilized during retrieval. To validate this hypothesis, they propose LABCLIP, a simple method that applies a lightweight linear transformation to the text embeddings. This approach significantly improves performance on compositional benchmarks (ARO, SugarCrepe, COCO) without retraining the frozen CLIP encoders, offering a practical and efficient solution for enhancing existing CLIP-based systems.

**Strengths:**

- The use of multiple, complementary experimental paradigms (linear probing, robustness to object count, conjunctive search) provides multi-faceted evidence for the uni-modal binding claim.
- The paper is well-written and clear. The core insight is presented early and reinforced throughout. Figures 1, 3, and 5 are particularly effective in illustrating the key concepts.

**Weaknesses:**

1. The conclusion that “the problem lies in cross-modal alignment” remains descriptive. The paper lacks a deeper theoretical or mathematical explanation of why the alignment loss fails to preserve binding (e.g., analyzing the contrastive objective’s geometry or the modality gap quantitatively).
2. The paper strongly asserts that the problem is only alignment. However, linear probing showing high accuracy only proves the information is linearly recoverable, not that it is represented in a way that is semantically robust or generalizable. High-dimensional embeddings can sometimes allow linear separation of concepts without the model having a true understanding of their binding.
3. The work focuses exclusively on attribute-object binding. While this is a fundamental primitive, true compositionality also involves relations, quantifiers, and negation. The paper does not demonstrate that the alignment fix would generalize to these more complex structures, which may require non-linear solutions.
4. The method for generating hard negatives on COCO relies on the heuristic from NegCLIP, which shuffles adjectives and nouns. This can create grammatically or semantically implausible captions (e.g., "a helmet man"). This risks teaching the model spurious correlations based on surface-level word co-occurrence rather than true structural binding, potentially inflating the reported gains.
5. The paper convincingly shows a linear layer is sufficient, but it does not compare against other lightweight, non-linear adapter methods (e.g., LoRA, small MLPs). Such a comparison would strengthen the claim that linearity is indeed the optimal and simplest solution. The argument that "linear is enough" is practical but not theoretically definitive.
6. As shown in Table 10 (Appendix), LABCLIP reduces zero-shot accuracy on standard single-object classification datasets (CIFAR, ImageNet). While expected, this trade-off is significant for practitioners and should be discussed more prominently in the main text, as it highlights the specialization of the method.

**Questions:**

My major concerns are outlined in the "Weaknesses" part.

---

> ### Author Response · Authors · 2025-11-21
>
> We thank reviewer **dUnx** for the detailed feedback. We address each point below.
>
> **1-1. Lack of theoretical or mathematical explanation of why the alignment loss fails to preserve binding**
>
> The alignment loss fails to preserve binding because the captions in the training data rarely specify both objects and their attributes. The contrastive loss rewards matches based on shared object mentions, not on whether attributes pair with the correct objects. As a result, the model has little incentive to align attribute-object pairs across modalities. This is consistent with object bias reported in VLMs (Schrodi et al., 2025).
>
> **1-2. "(e.g., analyzing the contrastive objective’s geometry or the modality gap quantitatively)"**
>
> We already report modality gap results in Appendix A.6.2 of the submission. Our goal was to test whether applying a linear transformation to text embeddings with LABCLIP reduces the modality gap. The results show that it decreases the gap in three out of four datasets. This suggests that better alignment may relate to a smaller modality gap, although the effect differs across datasets.
>
> **2. "Linear probing [...] only proves the information is linearly recoverable. High-dimensional embeddings can allow linear separation of concepts without [...] a true understanding of their binding"**
>
> Thank you for the comment. The linear separability we observe with probes cannot be explained by high dimensionality alone. The sanity check (lines 270–277) supports this. We train a randomly initialized CLIP model under a BoW constraint, where it learns to recognize attributes but not their associations with objects. If high dimensionality were sufficient, this encoder should also yield high probe accuracy. It does not: the probes fail to generalize on the test sets (0.66 for image, 0.85 for text vs. 0.96 for original CLIP). This shows that CLIP’s image and text embeddings contain additional structure that supports binding.
>
> **3. Generalization to more complex compositional structures (relations, quantifiers, and negation)**
>
> Thank you for the discussion. Indeed, our work is focused on the attribute-object binding, rather than other compositional issues like relations and negations. This is because the current CLIP variants still struggle severely with the attribute-object binding (Lewis et al., 2024). We tackle one problem at a time.
>
> Nevertheless, we have already shown in the submission that our method applies to other compositional structures. LABCLIP improves by **19%** over CLIP on the VG-Relations subset of ARO that includes relations between objects (Table 4).
>
> We also provide additional results for negation and spatial relations to assess how the method generalizes to these compositional structures.
>
> **Negation:**
> - We evaluate on the multiple-choice subsets of NegBench [b].
> - LABCLIP, which was trained only on attribute-object binding, does not generalize to negation data. It matches CLIP on COCO MCQ (**39.3** vs **39.3**) and VOC MCQ (**38.7** vs **38.7**).
> - For LABCLIP-neg, we use half of the NegBench subsets for training and the remaining half for testing. With this explicit negation supervision, accuracy increases to **70.1** on COCO MCQ and **81.4** on VOC MCQ.
>
> | Model | COCO MCQ | VOC MCQ |
> | - | - | - |
> | CLIP | 39.3 | 38.7 |
> | LABCLIP | 34.5 | 38.7 |
> | LABCLIP-neg | 70.1 | 81.4 |
>
> **Spatial relations:**
> - We evaluate on the What’sUp benchmark [c].
> - LABCLIP, trained only on attribute-object binding, performs similarly to CLIP (**31.4** vs **31.2** on What’sUp; **48.2** vs **47.3** on COCO-Spatial; **45.5** vs **47.3** on GQA-Spatial).
> - LABCLIP-spatial , with explicit spatial supervision (using half of each dataset for training and the rest for testing), improves across all datasets (**53.7**, **64.5**, **54.8**).
>
> | Model           | What'sUp | COCO-Spatial | GQA-Spatial |
> | --------------- | -------- | ------------ | ----------- |
> | CLIP            | 31.2     | 47.3         | 47.3        |
> | LABCLIP         | 31.4     | 48.2         | 45.5        |
> | LABCLIP-spatial | 53.7     | 64.5         | 54.8        |
>
> In summary, LABCLIP, trained only on attribute-object binding, does not generalize to negation or spatial relations. The same linear transformation approach works when given task-specific training data. It would be useful to first study how negation and spatial relations are represented in unimodal embeddings before applying linear or non-linear solutions.

---

> > ### Author Response · Authors · 2025-11-21
> >
> > **4. "The method for generating hard negatives [...]  can create grammatically or semantically implausible captions"**
> >
> > NegCLIP’s perturbation method shuffles adjectives within the adjective set and nouns within the noun set. This avoids cases like "helmet man," which involve two nouns. The swapped captions stay within the attribute-object pattern we target.
> >
> > **5. Comparison against non-linear adapters**
> >
> > Thank you for the suggestion. Non-linear adapters such as LoRA or small MLPs are indeed more expressive. However, our unimodal experiments show that attribute-object information is already *linearly separable* in the embeddings. We therefore focus on the minimal intervention needed to realign the two modalities, namely a linear transformation. Exploring non-linear adapters is an interesting direction, especially for more complex compositional structures.
> >
> >
> > **6. "LABCLIP reduces zero-shot accuracy [...] should be discussed more prominently in the main text"**
> >
> > Thank you for pointing this out. We will update the paper and add the discussion to the main text.
> >
> > **References:**
> >
> > [a] Schrodi et al. Two effects, one trigger: On the modality gap, object bias, and information imbalance in contrastive vision-language representation learning. ICLR 2025
> >
> > [b] Lewis et al. Does CLIP Bind Concepts? Probing Compositionality in Large Image Models. Findings of EACL 2024
> > Alhamoud et al. Vision-language models do not understand negation. CVPR 2025

---

### Author Response · Authors · 2025-12-03

We thank all the reviewers and the area chairs for their time and effort in assessing our work.

We provide this document to make it easier for the AC to follow the discussion, especially given the increased workload this cycle.

Below is the structure of this document:

- Summary of the work

- Summary of the reviews and discussion


	- 84zT

	- kJCN

	- dUnx

	- CZn4


- Timeline of the discussion

- Revisions and improvements

- Conclusion

---

## Summary of the work



Failures in compositional tasks in CLIP are widely reported, but it has been unclear whether they stem from the embeddings or from the alignment step. Our study clarifies this issue.



Our contributions are:

- Novel findings that CLIP's embeddings actually contain attribute-object binding signals in the respective image and text embeddings.

- Resulting conclusion that the BoWness reported in prior work arises from the cross-modal alignment step, not from limitations of the embeddings themselves.

- Practically useful observation that these binding signals can be recovered in existing pretrained representations through a simple linear transformation.


---

## Summary of the reviews



### 84zT (Score: 4 → 6)



**The reviewer asked about the differences between our work and [a,b].**

- Work [a] evaluates not only binding but also compositional generalization. [b] evaluates spatial reasoning, which is harder than attribute-object binding.

- Our work complements [a, b] by showing that the binding failures they observe come from cross-modal misalignment, not from limitations in the unimodal embeddings.

- Our experiments do not contradict their findings. In [a], models struggle to generalize to unseen attribute-object pairs. In [b], spatial reasoning remains far below human accuracy.




The reviewer found the additional explanations and experiments “insightful” and _increased_ the score from 4 to 6.



We added the suggested comparisons in the revision (§5.2, A.5) to better position our work.



[a] Lewis et al. Does CLIP Bind Concepts? Probing Compositionality in Large Image Models. EACL 2024

[b] Kamath et al. What's "up" with VLMs? Investigating their struggle with spatial reasoning. EMNLP 2024


---




### kJCN (Score: 4)



The reviewer raised the following points:



**What is the scope of our work with respect to other compositional tasks?**



We focus on attribute-object binding. Other complex compositional tasks (such as spatial relations, negation) are out of our scope.




**Does LABCLIP generalize to more complex compositional tasks?**



We provided results on negation and spatial relations. According to the new results, LABCLIP trained only on binding does not improve on these tasks, but it improves when given hard negatives specific to each task. This is expected, since binding, negation, and spatial relations each reflect a different aspect of compositionality.




**How does LABCLIP affect downstream performance?**



The downstream results were already in the paper (§A.5). They suggest a trade-off between single-object classification and attribute-object binding, similar to NegCLIP. To provide broader evidence for the impact of LABCLIP, we additionally trained LABCLIP on a dataset unrelated to COCO: CC3M. The results show a smaller impact on downstream tasks while maintaining consistently high performance on binding tasks (Table 11).






**Summary of the discussion**

Our initial response clarified several of the reviewer’s points, and they subsequently requested a more rigorous evaluation of the method during the discussion.



The reviewer hypothesized that **improvements on compositional benchmarks might come from an overlap between COCO (training) and ARO/SugarCrepe (evaluation), not from improved binding.**

- We trained LABCLIP on CC3M. LABCLIP-CC3M still outperforms CLIP.

- This confirms that the gains come from improved attribute-object binding.




Unfortunately, the discussion closed before we could present thу new experiments and discuss it further. These results are now included in our response and the revised paper.



Overall, we clarified the scope of our work, provided results on additional compositional tasks, clarified downstream performance, and added a more rigorous experimental setup beyond COCO to the revision. Together, we hope these additions address the points raised by the reviewer.

---

> ### Author Response · Authors · 2025-12-03
>
> ### dUnx (Score: 4)
>
> The reviewer raised the following concerns:
>
> **Our work lacks an explanation of why alignment loss fails to preserve binding**
>
> Captions in the training data rarely specify both objects and their attributes, leaving the alignment loss with insufficient signal to enforce cross-modal binding. This reflects a limitation of the data rather than the optimization.
>
> **Because the embeddings are high-dimensional, our linear-probe evaluation might show separability even if the embeddings do not encode binding.**
>
> We already show in §4.1 that BoW embeddings, even in high dimensions (768), do not allow linear probes to recover binding. Therefore, our evaluation does measure the degree of binding information in representations.
>
> **Does LABCLIP generalize to more complex compositional tasks?**
>
> We provided results on negation and spatial relations. According to the new results, LABCLIP trained only on binding does not improve on these tasks, but it improves when given hard negatives specific to each task. This is expected, since binding, negation, and spatial relations each reflect a different aspect of compositionality.
>
> **When adjective and nouns are shuffled as in NegCLIP, hard negative captions may include implausible examples such as ‘helmet man’.**
>
> We clarified that adjectives are shuffled only within adjective sets and nouns within noun sets, so no examples like 'helmet man' may occur.
>
> **Our work does not compare LABCLIP to non-linear adapters.**
>
> We use a linear layer because binding signals are already linearly separable. Non-linear adapters are more expressive but they would not isolate whether gains come from aligning the existing linearly recoverable signals or from adding capacity.
>
> **LABCLIP's zero-shot accuracy should be discussed in the main text.**
>
> We added this discussion in §5.2.
>
> **Summary of the discussion**
>
> Unfortunately, the reviewer did not respond during the discussion period.
>
> We addressed all of their concerns about alignment loss, generalization, non-linear adapters, high-dimensional separability, hard-negative captions, and downstream performance through added explanations, experiments, and revisions.
>
> ---
>
> ### CZn4 (Score: 8)
>
> The reviewer finds our contribution "highly relevant to the community".
>
> **The reviewer asked for more explanation of why BoWness emerges.**
>
> Prior work links this behavior to object bias and information imbalance in the training data, which we now make clearer in the revision (§5.3) [c].
>
> **The reviewer also asked about the role of the modality gap, since they originally suspected it to be the cause of cross-modal misalignment.**
>
> We clarify in the revision (§2) that the modality gap is itself a by-product of information imbalance between images and captions, and not the source of BoWness [c].
>
> **The reviewer provided suggestions on improving readability**, which we included in the revision.
>
> **During the discussion**, the reviewer did not respond further but was very positive about the paper in the initial review.
>
> We clarified the origins of BoWness and its relation to the modality gap, answered all questions, and improved readability. We hope these address all the reviewer’s comments.
>
> [c] Schrodi et al. Two effects, one trigger: On the modality gap, object bias, and information imbalance in contrastive vision-language representation learning. ICLR 2025
>
> ---
>
> ## Timeline of the discussion
>
> - 12 Nov 2025. Reviews released to authors.
>
> - 20 Nov 2025. The authors posted a response.
>
> - 25 Nov 2025. Reviewer 84zT appreciated the additional explanations and experiments and increased the score (4 $\rightarrow$ 6).
>
> - 27 Nov 2025. Reviewer kJCN requested a more rigorous experimental evaluation.
>
> - 21-28 Nov 2025. Other reviewers did not engage in the discussion.
>
> - 28 Nov 2025. The discussion came to a halt, following the emergency PC decision.
>
> ---
>
> ## Revisions and improvements
>
>
> We revised the submission with additions highlighted in blue. Key updates include:
>
>
>
> - LABCLIP trained on CC3M in §5.2 (kJCN)
>
> - Comparison to the similar CLEVR dataset in §5.2 and Appendix A.5 (84zT)
>
> - Evaluation on spatial reasoning in §5.2 and Appendix A.5 (84zT)
>
> - Positioning our work among other benchmark and compositional tasks in Related work (kJCN, 84zT, dUnx)
>
> - Discussion on downstream performance in §5.2 (dUnx)
>
> - Additional notes on the modality gap in Related work (CZn4)
>
> - Drawn a box to highlight incongruency in Figure 5 (84zT)
>
> - Citation correction at Line 323 (84zT)
>
> - Number of options added in §3.3 and §4.1 (CZn4)
>
> - Benchmark names added in Related Work for readability (CZn4)
>
> ---
>
> ## Conclusion
>
> The paper has improved substantially following the reviews. We clarified the scope, added explanatory notes, compared our work to others, and included more rigorous evaluations.
>
> We sincerely appreciate the reviewers’ effort and thoughtful feedback, which has been invaluable in strengthening the work.

---

### Meta-Review · Area_Chair_gKKz · 2026-01-07

**Summary:**

Reviewers broadly agreed that the paper makes a clear and valuable empirical contribution by isolating the source of CLIP’s bag-of-words behavior to cross-modal alignment rather than missing unimodal binding information. The experiments are carefully designed, and the rebuttal addressed several key concerns, including clarification of differences from prior work, mitigation of training–evaluation overlap via CC3M experiments, and clearer discussion of downstream trade-offs.

Some concerns remain, mainly the lack of a deeper theoretical explanation of why the contrastive objective leads to cross-modal BoW behavior and the limited generalization beyond attribute–object binding without task-specific supervision. However, these issues are well-scoped and do not undermine the main empirical findings. Overall, the strengths, clarity, and practical relevance outweigh the remaining gaps, supporting an accept (poster) decision.

**Reviewer Concerns:**

**Reviewer Concerns**

* **84zT:**
  The main concern about how this work relates to prior binding and spatial reasoning papers was addressed. The authors clarified the distinction between unimodal binding vs. cross-modal alignment, added evaluations on the relevant datasets, and explained the shuffled split. This appears to have resolved the reviewer’s key doubts.

* **kJCN:**
  The concern about training–evaluation overlap and COCO-specific bias was addressed by additional experiments training on CC3M, which largely preserve the gains. The scope limitation to attribute–object binding is now clearly stated, and downstream trade-offs are discussed. While the method does not generalize automatically to harder compositional tasks, this is acknowledged rather than claimed away.

* **dUnx:**
  Several empirical concerns were addressed (high-dimensional linear separability, hard-negative construction, downstream performance visibility). However, the core concern about the lack of a deeper theoretical or formal explanation for why the contrastive objective induces cross-modal BoW behavior remains outstanding, as does the absence of comparisons to non-linear adapters.

* **CZn4:**
  Minor clarity and positioning issues were addressed in the revision. The broader “why” question behind BoWness is discussed more clearly, but remains largely descriptive rather than theoretically resolved, which is acceptable given the paper’s empirical focus.

**Reviewer Scores:**

Reviewer 84zT: 4 → 6 (reviewer explicitly said they would bump the score)

Reviewer kJCN: 4 → 6 (authors added a more rigorous setting via CC3M; this directly addresses the reviewer’s main concern and reasonably merits +2)

Reviewer dUnx: 4 → 4 (major theoretical/accounting concerns remain; no clear signal they would change)

Reviewer CZn4: 8 → 8 (already strongly positive; no indication of change)

Average (post-discussion): 6.0

---

### Decision · Program_Chairs · 2026-01-26

Accept (Poster)